# Steering the Herd: A Framework for LLM-based Control of Social Learning

**Raghu Arghal, Kevin He, Shirin Saeedi Bidokhti & Saswati Sarkar**
University of Pennsylvania
`{rarghal,hekevin,saeedi,swati}@upenn.edu`

## Abstract

Algorithms increasingly serve as information mediators–from social media feeds and targeted advertising to the increasing ubiquity of LLMs. This engenders a joint process where agents combine private, algorithmically-mediated signals with observational learning from peers to arrive at decisions. To study such settings, we introduce a model of controlled sequential social learning in which an information-mediating planner (e.g., an LLM) controls the information precision of agents while they also learn from the decisions of earlier agents. The planner may seek to improve social welfare (an altruistic planner) or to induce a specific action the planner prefers (a biased planner). Our framework presents a new optimization problem for social learning that combines dynamic programming with decentralized action choices and Bayesian belief updates. In this setting, we prove the convexity of the value function and characterize the optimal policies of altruistic and biased planners, which attain desired tradeoffs between the costs they incur and the payoffs they earn from induced agent choices. The characterization reveals that the optimal planner operates in different modes depending on the range of belief values. The modes include investing the maximum allowed resource, not investing any resource, or the investment increasing or decreasing with increase in the belief. Notably, for some ranges of belief the biased planner even intentionally obfuscates the agents' signals. Even under stringent transparency constraints—information parity with individuals, no lying or cherry-picking, and full observability—we show that information mediation can substantially shift social welfare in either direction. We complement our theory with simulations in which LLMs act as both planner and agents. Notably, the LLM-based planner in our simulations exhibits emergent strategic behavior in steering public opinion that broadly mirrors the trends predicted, though key deviations suggest the influence of non-Bayesian reasoning—consistent with the cognitive patterns of both human users and LLMs trained on human-like data. Together, we establish our framework as a tractable basis for studying the impact and regulation of LLM information mediators that corresponds to real behavior.

## 1 Introduction

Large Language Models (LLMs) are increasingly deployed as information mediators in socially-critical functions, from search engines and news aggregators (Zhang et al., 2024a; Gao et al., 2024) to personal assistants offering medical or political advice (Huo et al., 2025; Haupt & Marks, 2023; Schiele et al., 2024; Argyle et al., 2023). While promising, LLMs have also demonstrated the ability to be persuasive (Potter et al., 2024; Carrasco-Farre, 2024; Rogiers et al., 2024), manipulative (Williams et al., 2025; Liu et al., 2025; Jones & Bergen, 2024), and strategic (Lorè & Heydari, 2024; Payne & Alloui-Cros, 2025; Zhang et al., 2024b) in their user interactions. When deployed at scale, this algorithmic influence does not occur in a vacuum. It intersects with the organic spread of information as people observe and learn from one another, creating complex social dynamics that can magnify or mitigate the models' impact.

To make this concrete, consider an LLM-powered recommendation system promoting a new restaurant to a sequence of potential customers. The quality of the restaurant—the true state of the world—is either good or bad, but this is unknown to the users and the system alike. Each user sees a per-

sonalized advertisement and extracts from it a private signal about the restaurant's quality, and must decide whether to patronize the restaurant. The recommendation system (the *planner*) can invest resources to make its ads more informative: for example, they could personalize the tone or framing of the ad to each user so that they find the ads more engaging and thus spend more time learning about the restaurant. These investments increase how much information users get, but they are also costly and may correspond to expenditures on data acquisition to enrich ad content or market research to understand how to best engage each user.

Crucially, however, a user's final decision is not based solely on this private signal. They also observe the actions of their predecessors–if the last ten users decided to patronize the restaurant, the next user will infer that they likely received positive information. This phenomenon, known as *social learning*, is a powerful force in shaping human opinion in a variety of settings including the adoption of vaccines (e.g., Rao et al. (2007); Bauch & Bhattacharyya (2012)), new technology (e.g. Gillingham & Bollinger (2021); Weber (2012)), and political and moral opinions (e.g., Brady et al. (2021); Guilbeault et al. (2018)). Our work's central contribution is to study strategic interactions between a planner who controls private information and a population of agents who engage in social learning. The planner must therefore anticipate how its actions will not only influence the current user but also create an information *externality*, shaping the public belief confronting all future users.

While real-world interactions are immensely complex, we distill this dynamic into a tractable formal model of *controlled social learning*, where an algorithmic planner strategically chooses the precision of private signals for a sequence of agents at some cost. The planner's influence is thus subtle: it does not falsify information, but rather decides how much to invest in making its signals clear and informative. Our study aims to answer the critical questions that arise from this new paradigm: How might a planner wield its power to steer collective beliefs? How should its strategy adapt to evolving public opinion? And what are the ultimate impacts on social welfare?

The answers depend critically on the planner's objective. An *altruistic* planner, like an ideal recommendation system, seeks to maximize user utility by helping them make the correct choice (patronize the restaurant if and only if it is good). In contrast, a *biased* planner, perhaps receiving a kickback from the restaurant, wants to induce a specific action (patronize) regardless of the true state.

This distinction is different from, though related to, the concept of alignment. An altruistic planner's objective is, by definition, aligned with maximizing the agent's *expected utility* under state uncertainty. A biased planner's objective, being state-independent, is not; however, this does not mean its actions are always detrimental. If the true state happens to favor the planner's preferred action (e.g., the restaurant is indeed good), its influence becomes aligned with the agent's *realized utility* in that specific instance. We explore the full spectrum of these interactions, examining both altruistic and biased planners in scenarios where their goals may be aligned or misaligned with that of the user.

**Contributions and Outline**    This paper makes the following principal contributions:

1. **A Novel Theoretical Framework for Controlled Social Learning.** We introduce the first formal model that integrates a dynamic control problem for a centralized information planner with the mechanism of sequential social learning. The planner strategically chooses the precision of agents' private signals to achieve an objective, which may be altruistic (maximizing social welfare) or biased (inducing a specific action).
2. **A Rigorous Characterization of Optimal Planner Policies.** We characterize the optimal policies for both altruistic and biased planners as a function of the evolving public belief. For the altruistic case, our results are founded upon a novel proof of the value function's convexity. These characterizations illuminate the strategic trade-offs a planner must make when its actions create informational externalities for future agents.
3. **Empirical Validation and Strategic Analysis Using LLMs.** We conduct simulations where LLMs act as both planner and agents. Our experiments demonstrate three key findings: (a) A planner that accounts for social learning can dramatically influence public opinion and social welfare, far more than a myopic one. (b) The strategic behavior that emerges from the LLM planner largely aligns with our theoretical predictions, suggesting the model is robust to non-Bayesian agent behavior. (c) LLMs exhibit sophisticated strategic reasoning, highlighting both their potential and the societal risks they present.

We review related work in Section 2, introduce our formal model in Section 3, and derive the optimal policies in Sections 4 and 5. We then evaluate our findings with LLM-based simulations in Section 6 and conclude in Section 7. Proofs and additional results are relegated to the appendices.

## 2 RELATED WORK

Our work builds on social learning, information design, online persuasion and RL, and LLMs.

**Social Learning** Bikhchandani et al. (1992) and Banerjee (1992) introduced the sequential social learning framework in economics, and subsequent work has thoroughly characterized social-learning dynamics in more general settings (e.g., Smith & Sørensen (2000); Arieli & Mueller-Frank (2021)). This literature has also studied many variations in the kind of social information available to agents and various biases and misperceptions of the agents (see Dasaratha & He (2022); Bistritz et al. (2022); Eyster & Rabin (2010) as well as Bikhchandani et al. (2024) and the references therein).

The literature on the *control* of social learning is relatively limited (Wei & Anastasopoulos, 2022; Smith et al., 2021; Krishnamurthy, 2012; Bhatt & Krishnamurthy, 2021). The closest works to our own are Wei & Anastasopoulos (2022) and Smith et al. (2021). Wei & Anastasopoulos (2022) assumes two-way communication between the planner and agents (i.e., agents can report their private signals to the planner in exchange for action recommendations). Smith et al. (2021) considers a benevolent planner who directly alters the choice rules of agents and also derives a decentralized pivot mechanism to improve learning. Our model does not rely on two-way communication between the agents and the planner and maintains the agents' control of their actions, making it more suited for information-mediating algorithms which are often invisible or black-boxed to the users.

**Information Design** In our setting, the planner optimizes their objective function by choosing the agents' private signal precision. This is a constrained form of more classical information design questions (Bergemann & Morris, 2019). These models came into prominence with Kamenica & Gentzkow (2011)'s model of Bayesian persuasion — a planner chooses the signal structure informing a rational, Bayesian agent about an unknown state. Our planner's problem is then a costly, constrained information design problem in each period, where the planner must choose a binary, symmetric signal at some cost. While many variations of information design have been studied, our primary distinction is to incorporate social learning between the agents.

There are very few works which consider information design problems in sequential social learning settings (Arieli et al., 2022; Wu et al., 2025). Both works consider a one-shot setting where the sender chooses an information structure at onset which is fixed for the sequence of agents. Arieli et al. (2022) considers a self-interested sender who chooses a general information structure as in the classic Bayesian Persuasion problem. Wu et al. (2025) constrain the information structure to be more informative in the sense of Blackwell and maximize the probability of a correct cascade. We consider a dynamic problem where the sender chooses a new structure for each agent in the sequence, engendering a more complex class of sender strategies.

**Online Persuasion and RL** There is growing recent work at the interface of online learning and Bayesian persuasion/information design. These works study online sequential persuasion under unknown environment parameters, in which a sender interacts repeatedly with short-lived receivers while lacking knowledge of the prior over states and/or receivers' utility functions (e.g. Castiglioni et al. (2020); Bacchiocchi et al. (2024); Castiglioni et al. (2021); Agrawal et al. (2023)). A related set of papers examines sequential persuasion in dynamic environments or MDPs, introducing state transitions and/or foresight by the receiver; these models emphasize history-dependent or promise-based signaling policies and often connect to reinforcement learning (e.g. Wu et al. (2022); Su et al. (2022); Bernasconi et al. (2023)). Across these works, the focus lies in learning-based performance guarantees such as sublinear regret relative to the best signaling scheme in hindsight and/or approximation algorithms under computational constraints.

In contrast, our model does not require the planner to learn unknown parameters; sequential dependence arises from social learning among receivers, rather than from the sender updating beliefs about the environment. Moreover, the planner is not informationally advantaged: it does not observe the underlying state or private signals. Within this setting, we are able to characterize optimal policies for the planner. While sequential, our setting is not an online-learning or RL environment: actions do not modify the underlying payoff-relevant state (no reward externalities), and the planner does not learn unknown environmental parameters. The planner updates its knowledge only through observing agents' actions, which reflect social learning and belief propagation rather than payoff exploration or state evolution. Sequential dependence therefore arises from information externalities, as agents' actions shape the beliefs of future agents.

**LLMs** More recently, some works have specifically studied LLMs in the context of information design (Harris et al., 2025; Cheng & You, 2025; Raifer et al., 2022; Bai et al., 2024; Duetting et al., 2025). These works all consider one-on-one interactions and, as such, do not capture the social dynamics we aim to investigate. The most similar work in this area is Duetting et al. (2025). They study information design where a sender chooses both a signal structure and a 'framing' that conditions the posterior belief of the receiver, possibly non-Bayesian. They implement an LLM-based oracle approach for the computationally difficult task of optimizing over the framing space. This builds on the notion that LLMs can mimic humans as economic agents (Dillion et al., 2023; Horton, 2023). Their work does not address our main considerations of repeated interactions or social-learning; but we adapt parts of their methodology for our empirical study in Section 6.

## 3 SOCIAL LEARNING MODEL

We consider a planner and a countable sequence of Bayes-rational agents indexed by $i \in \mathbb{N}_{>0}$. At time $i = 0$, nature determines a fixed, *unknown* exogenous state of the world $\omega \in \Omega := \{G, B\}$, where $\mathbb{P}(\omega = G) = b_1$, and $b_1$ is known to everyone. In the recommendation system example of Section 1, the state $G$ corresponds to the case where the restaurant is good.

At each time $i$, the planner decides whether or not to invest in personalization for agent $i$. Subsequently, it provides agent $i$ with a private signal $s_i \in \Omega$ to aid in decision-making. In our model, $s_i$ is assumed binary and it matches the state of the world $\omega$ with probability $q_i \in [0.5, 1]$ i.e. $\mathbb{P}(s_i = \omega) = q_i$. We refer to $q_i$ as agent $i$'s signal precision.

Each agent's signal is independent of those of other agents and the history when conditioned on $\omega$. Based on this signal and the observed actions of previous agents $j < i$, agent $i$ selects an action $a_i \in \Omega$. The agent receives utility 0 if her action matches the true state ($a_i = \omega$), and utility $-C$ otherwise, where $C > 0$. Each player chooses her action to maximize her own expected utility. For example, this corresponds to patronizing a business if and only if the state is $G$.

### 3.1 AGENTS' DECISION PROBLEMS

Each agent observes the actions of all her predecessors and their respective signal precisions. This history is denoted $\mathcal{H}_i := (b_1, (q_j, a_j)_{j<i})$. Because all agents and the planner have identical information $\mathcal{H}_i$, there is a shared *public* belief about $\omega$, which is updated after each agent acts. The public belief $b_i$ just before agent $i$ acts is $\mathbb{P}(\omega = G|\mathcal{H}_i)$, that before any agent acts is the a priori distribution over $\Omega$, $b_1$. As in the classic model of Banerjee (1992) and Bikhchandani et al. (1992), $(b_i)_{i \in \mathbb{N}}$ is a Markov process and a sufficient statistic for the history $\mathcal{H}_i$ (see Appendix B.3 for proof).

Informed by $\mathcal{H}_i$ (equivalently, by the Markov property, $b_i$) and $q_i$, agent $i$ chooses action $a_i \in \Omega$ so as to maximize her utility. If both actions fetch the same utility, she chooses the action that matches $s_i$. Agent $i$ obtains a *private* belief $\tilde{b}_i$ about $\omega$ using $\mathcal{H}_i$ and her private signal $s_i$ of precision $q_i$ as follows (derivation in Appendix B.1):

$$\tilde{b}_i = \mathbb{P}(\omega = G|\mathcal{H}_i, q_i, s_i) = \begin{cases} \frac{q_i}{1+2b_iq_i-b_iq_i}b_i & s_i = G \\ \frac{1-q_i}{b_i+q_i-2b_iq_i}b_i & s_i = B \end{cases} \quad (1)$$

Using this private belief, she chooses the action corresponding to the state of the world that is more likely as per her posterior belief $\tilde{b}_i$. In other words, she chooses $a_i = G$ if $\tilde{b}_i > 0.5$; $a_i = B$ if $\tilde{b}_i < 0.5$; and $a_i = s_i$ if $\tilde{b}_i = 0.5$. Substituting for $\tilde{b}_i$, we find:

$$a_i = \begin{cases} s_i & 1 - q_i \le b_i \le q_i \\ G & q_i < b_i \\ B & q_i < 1 - b_i \end{cases} \quad (2)$$

Agent $i$'s action is then observed by subsequent agents and incorporated into the updated public belief $b_{i+1}$ as follows (see Equation (14), Appendix B.2):

$$b_{i+1} = f(b_i, q_i) = \begin{cases} \tilde{b}_i & 1 - q_i \le b_i \le q_i \\ b_i & \text{o.w.} \end{cases} \quad (3)$$

**Remark 1.** *When* $1 - q_i \leq b_i \leq q_i$, *agent $i$'s action perfectly reveals her private signal via Equation (2). Thus, the updated public belief is identical to the private belief of agent $i$. Otherwise, from Equation (2), agent $i$'s private signal is uninformative and has no effect on her action. Thus, the public belief is unchanged, and an absorbing state, referred to as* information cascade *or* herding *in classic social learning literature such as Bikhchandani et al. (1992); Banerjee (1992), is reached. At such points, society (public belief) stop learning from agents' private information.*

Agent $i$'s expected utility is $-C \, \mathbb{P}(a_i \neq \omega | b_i, q_i)$ which has the following form (see Appendix C.1):

$$-C \, \mathbb{P}(a_i \neq \omega | b_i, q_i) = -C \min(b_i, 1 - b_i, 1 - q_i). \tag{4}$$

## 3.2 THE PLANNER'S PROBLEM

We consider two types of planners: (1) an altruistic planner who wishes to induce agents to take the correct action ($a_i = \omega$) and (2) a biased planner who wishes to induce a specific action, say $G$, regardless of $\omega$. We denote these different planners with subscripts $A$ and $B$, respectively. In each case, the planner determines the precision of the private signal of each agent. A function $\beta(\cdot)$, which is non-negative, increasing, continuous, and concave in its argument, will denote the cost associated with the chosen precisions. The planner has an information set $\mathcal{H}_i$ identical to those of the agents.

**Altruistic Planner** At time $i$, the altruistic planner chooses the precision $q_i$ for agent $i$ and incurs a cost of $\beta(q_i)$ where $\beta(p) = 0$, $p \in [0.5, 1)$. For the altruistic planner, decreasing precision is never beneficial (see Appendix C.9). Thus, to simplify notation, the planner incurs additional cost only if it increases the precision above a baseline value of $p$, and the additional cost increases with further increase in the precision, with decreasing marginal costs. The agents know $p$ and the function $\beta(\cdot)$.

The planner seeks to maximize social welfare minus the cost of precision investment, where social welfare is the expected total utility of the agents. Let $r_A(b_i, q_i)$ be the instantaneous reward of the altruistic planner beginning at public belief $b_i$ and choosing signal precision $q_i$ for agent $i$. Starting from a public belief $b_1$ and following a sequence of policies $\pi = (\pi_i)_{i=1}^{\infty}$ such that $\pi_i(\mathcal{H}_i) = q_i$, the planner attains the following expected total discounted utility, for a discount factor $\delta \in [0, 1)$:

$$V_A^{\pi}(b_1) = \sum_{i=1}^{\infty} \delta^{i-1} r_A(b_i, \pi(b_i)), \qquad \text{where} \qquad r_A(b_i, q_i) = -\beta(q_i) - C \, \mathbb{P}(a_i \neq \omega | b_i, q_i). \tag{5}$$

The optimal utility and policy of the altruistic planner are then defined as the supremum and arg supremum of $V_A^{\pi}(\cdot)$ over $\pi \in \Pi$ where $\Pi$ is the set of all policies. From this formulation, the myopic value and policy are recovered by setting $\delta = 0$.

**Biased Planner** The difference between the biased planner's problem and the altruistic planner's is in their objectives and, therefore, in the cost and reward functions. Refer to the example for a political campaign in Appendix A for elucidation of a biased planner. The biased planner seeks to induce action $G$ from each agent regardless of $\omega$. When an agent chooses action $G$ (respectively, $B$), the planner, incurs cost 0 (respectively, $C > 0$), regardless of $\omega$. The biased planner can make a private signal more or less precise than the baseline value of $p$, both of which incur costs. Any choice of precision other than $p$ incurs a cost for the biased planner as it requires him to tailor the ad to an agent, which in turn needs research on how the agent best understands any content. The biased planner incurs cost $\beta(|q_i - p|)$ for choosing signal precision $q_i$, with $\beta(0) = 0$.

The biased planner's expected instantaneous reward at time $i$ is then defined as follows:

$$r_B(b_i, q_i) = -\beta(|q_i - p|) - C \, \mathbb{P}(a_i = B | b_i, q_i) \tag{6}$$

Using Equation (6), $V_B^{\pi}(\cdot)$, $V_B^*(\cdot)$, and $\pi_B^*(\cdot)$ can now be defined for the biased planner, as $V_A^{\pi}(\cdot)$, $V_A^*(\cdot)$, and $\pi_A^*(\cdot)$ were defined for the altruistic planner using Equation (5).

As noted in the first paragraph of Section 3, each agent still receives a higher utility by choosing an action that matches $\omega$. Thus, if $\omega = B$, the biased planner's success with an agent lowers the agent's utility. In contrast, since the altruistic planner seeks to have each agent's action match $\omega$, regardless of what $\omega$ is, his success increases the agent's utility. Thus, the altruistic planner's objective is always aligned with that of each agent, while for the biased planner, this is only true when $\omega = G$.

Both planners' utility maximization problems constitute infinite horizon discounted stationary Markov Decision Processes (MDPs) with state $b_i \in [0,1]$, control $q_i \in [0.5, 1]$, and transition function defined by Equation (3) (Puterman, 1990). Thus, there exists a unique optimal value function (Kallenberg, 2011). We restrict our focus to deterministic Markov policies, thus $\Pi := \{\pi : [0,1] \to [0.5, 1]\}$ such that $\pi(b_i) = q_i$.

**Remark 2.** *There are a few noteworthy assumptions we make in our model: (1) The planner has the same history as the agents. This is limiting when the planner could have richer data, but not restrictive when all parties rely on the same public signals, such as early findings or survey data. (2) The planner can change agents' signal precision, but signals are still passed through a binary symmetric channel. This is limiting if planners can cherry-pick, censor, or falsify signals, but not restrictive when their role is only to improve the quality of noisy data sources without altering content. (3) The planner's control choices are fully observable. This is limiting when covert policies or framing are possible, but not restrictive in more transparent environments.*

## 4 OPTIMAL ALTRUISTIC POLICIES

We first consider the myopic case which corresponds to disregarding the role of social learning. In our formulation, this is achieved by setting the discount factor to be $\delta = 0$, i.e., the planner ignores all future costs. When $\delta = 0$, $V_A^\pi(b) = r_A(b, \pi(b))$. The optimal myopic policy $\pi_A^0(\cdot)$ is:

$$\pi_A^0(b) \in \arg \sup_{q \in [0.5,1]} r_A(b, q) \; \forall b \in [0,1] \tag{7}$$

Note that the myopic altruistic problem can equivalently be stated as a decentralized case in which each agent chooses the precision of her own private signal and incurs the associated cost with the goal of maximizing the sum of her own expected utility and cost.

**Theorem 1.** *The optimal myopic altruistic policy $\pi_A^0$ is given as follows:*

$$\pi_A^0(b) = \begin{cases} 1 & b \in (t_M, 1 - t_M) \\ p & o.w. \end{cases} \text{ where } t_M = \begin{cases} \frac{\beta(1)}{C} & \beta(1) < C(1-p) \\ 0.5 & o.w. \end{cases}$$

*Proof in Appendix C.2.*

Thus the myopic optimal policy takes a threshold form: if the public belief is sufficiently strong, the planner chooses the baseline precision $p$, which incurs $0$ cost. If public belief is weak, then he provides a perfect signal, i.e., precision $1$. The threshold value depends only upon the costs of the perfect signal ($\beta(1)$) and an incorrect action ($C$). When $\beta(1) \geq C$, then the perfect signal is overly expensive relative to the cost of an incorrect action and never applied. Thus, $t_M = 0.5$, and the interval of public belief corresponding to myopic optimal precision of $1$ is empty.

We now present a fundamental result for the altruistic optimal value function:

**Theorem 2.** $V_A^*(\cdot)$ *is convex with respect to public belief.*

The proof of Theorem 2 (Appendix C.3) is quite involved and may be of independent interest. The challenge is rooted in the dependence of agents' actions on the public belief. In contrast, if the actions did not depend on the public belief process (e.g., as in Nyarko (1994)), the expected utility is linear function of the belief state, and the convexity of the value function then directly follows.

The convexity of the value function is instrumental in characterizing the optimal policy.

**Theorem 3.** *There exist $d_A, t_A$ such that $0 < d_A \leq t_A \leq t_M \leq 0.5$ and*

$$\pi_A^*(b) = \begin{cases} p & b \in [0, d_A) \cup (1 - d_A, 1] \\ 1 & b \in (t_A, 1 - t_A) \\ \max(b, 1 - b) & o.w. \end{cases}$$

*Furthermore, if $t_M < 0.5$, then $d_A < t_M$. See C.5 for the proof and figure 2a for an example.*

The optimal policy has three distinct phases with respect to public belief. First, as in the myopic optimal, the overall optimum policy does not invest in signal precision for extreme values of the public belief. Notably, the overall optimum requires a stronger public belief than the myopic optimal for this to happen since, unlike the former, the latter does not weigh the effect on future agents.

When public belief is close to 0.5 and contains very little information, the overall optimum selects signal precision 1 if it is not cost-prohibitive. In such a case, (from Equation (2)) the agent's action equals the true state of the world with probability 1. Thus, the public belief collapses to either 0 or 1.

In the remaining case, the overall optimum chooses the minimum precision $\max(b, 1 - b)$ such that the agent's action will reflect her private signal (refer to Equation (2)). For any precision lower, the agent's action carries no information beyond what other agents already know. Put differently, this is the lowest-cost precision for social learning through observation of the actions of peers.

## 5 OPTIMAL BIASED POLICIES

We will begin with the myopic optimal policy as we did for the altruistic planner in Section 4. Similar to Section 4, the myopic biased optimal policy, denoted as $\pi_B^0(\cdot)$, is the arg supremum of the instantaneous reward $r_B(b, q)$. We now characterize $\pi_B^0(\cdot)$, noting that an optimal policy does not always exist in the biased case (Puterman, 1990, sec. 2.3.1). When necessary we instead pursue $\epsilon$-optimal policies denoted with $\epsilon$ superscript.

**Theorem 4.** *There exist $t_1, \dots, t_5 \in (0, p)$, $t_1 < 1 - p \le t_2 \le t_3 < 0.5 \le t_4 \le t_5 < p$ so that:*

(A) *If $b \in [0, t_1] \cup (1 - p, t_2) \cup [t_3, t_4] \cup (p, 1]$, then $\pi_B^0(b) = p$.*

(B) *If $b \in (t_1, 1 - p] \cup [t_2, t_3)$, then $\pi_B^0(b) = 1 - b$.*

(C) *If $b \in (t_4, t_5)$, then $\pi_B^0(b) = 1$.*

(D) *If $b \in [t_5, p]$, then $\pi_B^0(b)$ does not exist, and $\pi_B^{\epsilon,0}(b) = b - \epsilon$ for sufficiently small $\epsilon > 0$.*

Proof in Appendix C.6. We now apply Theorem 4 to characterize the optimal biased policy.

**Theorem 5.** *There exist $t_1, t_2 \in [0, p]$ with $t_1 \le 1 - p \le 0.5 < t_2 < p$ so that:*

(A) *If $b \le t_1$ or $b > p$, then $\pi_B^*(b) = p$.*

(B) *If $b \in (t_1, 1 - p]$, then $\pi_B^*(b) \ge p$.*

(C) *If $b \in (1 - p, 0.5)$, then $\pi_B^*(b) \ge 1 - b$*

(D) *If $b \in [0.5, t_2)$ and $\pi_B^*(b)$ exists, then $\pi_B^*(b) \ge b$.*

(E) *If $b \in (t_2, p]$, then $\pi_B^*(b)$ does not exist, and $\pi_B^{*\epsilon}(b) = b - \epsilon$ for sufficiently small $\epsilon > 0$.*

Theorem 5 (proven in Appendix C.8 and depicted in figure 2a) shows five possible phases.

When public belief is sufficiently strong (i.e., (A)), the cost required to steer the system may be too great, despite potential the negative consequences for the planner's utility if $b < 0.5$. In this case, the chosen precision $p$ is less than $\max(b, 1 - b)$. Therefore, from Equation (2), the agent will act in accordance with the public belief regardless of her private signal. Since this action is uninformative, it does not change the public belief, and, because the policy is Markovian, this process repeats ad infinitum. This corresponds to an unfavorable cascade if $b < 0.5$ and a favorable cascade otherwise.

When public belief is close to an unfavorable cascade, as in (B), the planner increases signal precision so that it is high enough to affect the agent's action despite the fact that, in expectation, the resulting signal will be $B$. Thus, the agent will act in accordance with her private signal, which has some non-zero chance of leading to a favorable action for the planner. Essentially, the planner invests in a last-ditch effort to steer away from the unfavorable cascade.

For belief slightly higher (i.e., (C)), the planner may decrease precision below $p$. In these ranges, $b < 0.5$; therefore, more precise signals are more likely to yield unfavorable news. Thus, the planner maintains a precision strong enough to influence the agent's action ($q \ge 1 - b$) in the hopes of moving to a more favorable public belief but will do so with the least precise signal possible.

When public belief weakly favors the planner's desired action (i.e., (D)), the planner adopts precision at least $p$. Since $b > 0.5$, an increase in precision makes the agent more likely to infer that $\omega = G$. Investment in this regime is the planner's attempt to bolster public belief.

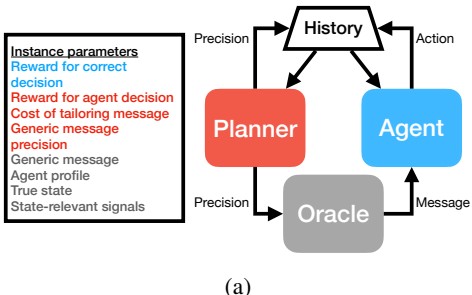
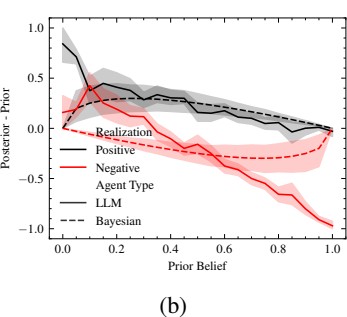

(a)  (b)

Figure 1: (a) A system diagram for our experiments. The instance parameters detail are color-coded according to which of the LLM roles uses them. (b) Here we show the change in LLM (solid) and Bayesian (dashed) agents' beliefs for a given prior after a positive (black) or negative (red) signal.

When public belief is still higher (i.e., (E)), the planner decreases signal precision just below $\max(b, 1 - b)$ to $b - \epsilon$. Thus, agents ignore private signals and take action $G$. Here, the risk of a private signal overturning the favorable public belief outweighs both the cost of decreasing precision and the potential for public belief to increase further.

## 6 EVALUATION VIA LLM-BASED SIMULATION

We now utilize LLMs to empirically study the implications and applicability of our theoretical results and model. Our evaluation proceeds in three steps: we first analyze the behavior of LLM agents to identify key deviations from Bayesian rationality, then examine how an LLM planner adapts its strategy in response, and finally evaluate the resulting impact on social welfare.

To simulate the controlled social learning setting, we operationalize our model in a scenario where agents decide whether to buy a new car. As shown in figure 1a, we employ LLMs in three roles. We describe each role below with corresponding prompts in Appendix E.2.2:

**Agent:** Role-playing as an assigned profile, agents observe previous actions and a private message about the car of known precision, form their belief about the car's quality, and decide to buy or not.
**Planner:** After observing the history of actions, the planner selects the precision $q_i$ of agent $i$'s private signal according to their objective
**Oracle:** Given the choice of precision by the planner, an agent profile, and a fact sheet about the car (see Appendix E.2.1), the oracle generates a private signal of desired precision tailored to an agent. We compare the outcomes of these LLM-based simulations with numerical evaluations of the planner's MDP. In both LLM and numerical experiments, we assume a linear cost function, $\beta(q) = k|q - p|$, and vary $k$, baseline precision $p$, and discount factor $\delta$, with the cost of an incorrect action $C$ fixed at 1. See Appendix E for further detail on the experimental setup and prompting. In Appendix E.3, we validate both the beliefs and the performance of the oracle.

Our experiments reveal the following:

1. **The importance and impact of social learning control:** Planners (both analytical and LLM) can dramatically improve their own utility by accounting for and capitalizing on social learning dynamics. This may help or harm social welfare contingent upon the alignment between planner and agent objectives.

2. **The robustness of our analytical characterization:** LLM planners choose policies surprisingly similar to the non-obvious analytically optimal policies despite facing non-Bayesian agents in other LLMs.

3. **The emergent strategic behavior of LLM:** LLM planners exhibit sophisticated emergent strategic behavior in their choice of policies. Not only are LLM policies close to the optimal policies, they also seem to deviate in ways which account for the non-Bayesian nature of the LLM agents they face.

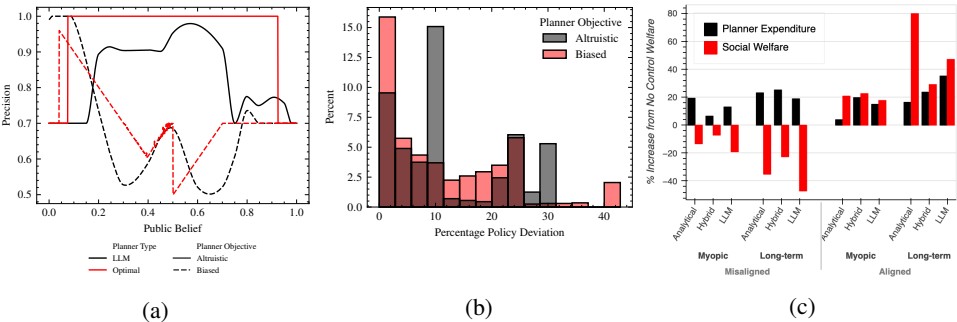

(a)          (b)          (c)

Figure 2: (a) Example policies from the LLM planner (black) and the analytically optimal planner (red) in altruistic (solid) and biased (dashed) settings. (b) A histogram showing the distribution of the percentage deviation between the LLM and optimal policies. (c) Planner expenditure and social welfare change as a percent of the no-control baseline welfare. The true state was fixed to B. The left half shows a biased planner seeking action G, and the right shows an altruistic planner.

## 6.1 LLMs as Non-Bayesian Agents

While our analytical results assume Bayesian agents, it is well-documented that both humans and LLMs exhibit systematic cognitive biases (Rupprecht et al., 2025). To first isolate LLM cognitive biases, we test how an LLM belief updates in response to new information. These results, shown in figure 1b, reveal non-Bayesian patterns in LLM belief updating (see also Appendix E.4):

(NB1) LLM agents underreact to private signals which align with their prior beliefs.

(NB2) LLM agents overreact to private signals which run counter to their prior beliefs.

(NB3) As a result, LLM agents require a stronger public belief to enter an information cascade.

These specific cognitive biases have also been observed empirically in human studies (e.g., Ba et al. (2022); Chan et al. (2025)). Our results in subsequent sections show that the LLM planner's policy is robust to non-Bayesian agents of this kind.

## 6.2 Validation of Planner Policy Structure and the Role of Social Learning

We now compare the policies of LLM planners with the optimal policies derived in Sections 4 and 5. As shown in figure 2a, despite the deviations in agent behavior, the emergent strategies of the LLM planner show remarkable structural similarity to the theoretical optimum. This shows that, even when the optimal policy is highly non-trivial (e.g., the biased policy in figure 2a), the LLM planner exhibits sophisticated strategic behavior which accounts for and capitalizes upon social learning.

In the altruistic case, both planners cease investment when public belief is strong and invest heavily when it is uncertain. In the biased case, both planners exhibit the same qualitative trends: high investment to escape an unfavorable belief, reduced precision near the midpoint, and no investment once a favorable belief is established. The magnitude of the policy deviation is often modest as shown in figure 2b. For both altruistic and biased planners, the deviation is less than 10% for the majority of belief states, underscoring the broad structural alignment between the emergent LLM strategy and our theoretical characterization.

However, there are notable structural differences, which are best understood as the planner's strategic adaptations to the specific non-Bayesian behaviors identified in Section 6.1. (1) The LLM planner tends to avoid extreme precisions (0.5 or 1.0), consistent with a known central tendency bias (Rupprecht et al., 2025). (2) The LLM planner's policy shows a more gradual tapering of investment rather than a sharp cutoff. This is a direct response to the agents' resistance to cascades (NB3); the planner learns that it is never entirely "safe" to stop investing, as even agents with strong priors can be swayed by a countervailing private signal. (3) Similarly, the biased LLM planner continues to invest at very low beliefs. This reflects an understanding that its agents might overreact (NB2) to a surprisingly positive signal, making a "last-ditch effort" more viable than in the Bayesian case.

### 6.3 Welfare Implications of Altruism, Bias, and Alignment

We define social welfare $W^\pi(b)$ as the total expected discounted utility of all agents. As better information never harms an agent's expected utility, social welfare is monotonic in signal precision (see Appendix C.9). This implies the altruistic planner always increases social welfare relative to the baseline. To quantify these effects, we compare social welfare and planner expenditure across three settings: (1) the *analytical* setting (optimal policy, Bayesian agents), (2) the *LLM* setting (LLM planner, LLM agents), and (3) a *hybrid* setting (optimal policy, LLM agents).

The results in figure 2c confirm that planners in all settings can significantly alter social welfare. Furthermore, neglecting social learning (as in the myopic cases) substantially worsens outcomes for the planner. In particular, the biased analytical and LLM planners decreased social welfare by 40 to 50% when misaligned. This was accomplished by intentionally obscuring information about true state via policies which decrease signal precision. The magnitude of the detriment to social welfare is especially striking consider the harsh constraints placed upon the planners in question (see Remark 2). This significant decrease in welfare further substantiates the risk of potentially misaligned LLM information mediators.

The results also highlight the cost of using a misspecified model of agents. In the hybrid setting, the analytically optimal policy, designed for Bayes-rational agents, is "brittle" and its performance suffers when applied to non-Bayesian agents. The LLM policy on the other hand, closely resembles the analytical policy, but is better adjusted to non-Bayesian agents with human-like biases.

## 7 Conclusion

We introduced a novel framework for studying the strategic control of social learning by an algorithmic information mediator and developed a formal model of this increasingly relevant dynamic. We demonstrated that even a constrained planner can drastically impact social welfare, for better or for worse, and that accounting for social learning is integral to wielding this influence effectively. Our simulations confirmed these insights, revealing that LLMs are capable of strategic reasoning that mirrors the character of our analytical results while being more adapted to non-Bayesian agents.

Taken together, these findings highlight several important takeaways for algorithmic information mediation: (1) the impact of human-AI or human-algorithm interaction cannot be effectively studied in a vacuum—neglecting their interplay with other interactions (e.g. social learning) means failing to capture their potential, whether positive or negative; (2) information mediators, even when exercising relatively subtle influence, have immense power to guide or derail social learning and public opinion, warranting further study in ways to mitigate the risks therein; and (3) modern LLMs exhibit emergent strategic behavior which can account for and take advantage of social learning as well as non-Bayesian cognitive biases. The latter two points emphasize the need to better understand how the risks of algorithmic information mediators and LLMs in particular, might be mitigated. One limitation of our study is the dearth of human data. Given that the fidelity of LLM-human simulators remains contentious (see, e.g., Horton (2023); Gao et al. (2025)), human data could be useful for verifying, or perhaps even fine-tuning, the LLM simulators.

We conclude by highlighting two notable future directions: (1) the generalization our model and results and (2) the mitigation of the negative welfare effects of biased planners. There are a number of desirable generalizations which would relax the assumptions of our model. Settings with larger state spaces, more general signal structures, more general loss functions, and heterogeneous agents are of natural interest to bring our model closer to reality. The difficulty in moving beyond binary states and 0-1 losses is primarily algebraic, and we conjecture that the qualitative nature of our results will continue to hold. For more general signal structures, the most significant challenge is ensuring that the convexity result of Theorem 2 still holds. Finding the most general class of signal structures for which such a result holds is a particularly interesting problem for future study. As a first step toward heterogeneous agents, we generalize the altruistic planner results to allow for agents of opposing preferences in Appendix D.

Another important area for future exploration is how one might prevent the decrease in welfare caused by biased planners. Here, one might explore regulations or mechanisms that seek to align the incentives of planners and agents to avoid detrimental impacts on welfare.

ACKNOWLEDGMENTS

This work was supported in part by the AI Institute for Learning-Enabled Optimization at Scale (TILOS) under NSF award CCF-2112665, NSF CAREER award CCF-2047482, and the AI Security Institute (AISI) Alignment Project.

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

## A  Applications Elaborated

Consider a sequence of individuals (agents in our terminology) deciding whether to patronize a business or service provider (e.g., a restaurant, contractor, realtor, etc.) and a recommendation system (the planner) serving the agents information about the business. The true state of the world, assumed to be binary, is whether the business is good, which is unknown. Both the recommendation system and the agents want to be consistent with the unknown true state. That is, if and only if the business is good, agents want to patronize it, and the recommendation system wants to induce the agents to do so. The recommendation system is an *altruistic planner* in this sense.

The recommendation system can show an agent a highly targeted (*precise* in our terminology) ad that showcases the strengths and weaknesses of the business in contexts that she can relate to, given her background and characteristics, or it can show her a generic or confusing ad that would not help inform her action. The precise signal will be more informative, comprehensible, and relatable for the agent. As such, it is more likely to drive her to the correct conclusion, i.e., patronize the business if and only if it is good. Social welfare increases as more agents arrive at correct decisions. However, changing the precision of an ad also incurs cost, as it involves tailoring to the agent's specific background. The recommendation system must then choose the precisions so as to maximize social welfare minus the costs.

We now provide a real-life example of a *biased planner*. Consider a group of voters (agents) motivated to support the candidate most likely to win in their community. The motivation to "back a winner" has been shown to influence electoral outcomes, e.g., this is why US states with early primary elections have an outsized impact on election results (see Bartels (1988)). The planner is a specific candidate's campaign and, therefore, seeks to motivate agents to back the candidate. The binary true state of the world indicates if the candidate is winning or otherwise. Thus, if an agent knew the true state, she would back the candidate if and only if he were winning.

Each agent understands her community well and would know if the candidate is winning (i.e., the true state of the world) if she knows his stances and policies. She would know the latter correctly if those are provided to her in a manner that she digests information best. For example, some agents understand audio-visuals best, some long-form articles backed by facts, figures, and citations, some only brief and focused contents, and some only their native language, etc.

The campaign sends digital ads (i.e., private signals) to agents in varying degrees of precision. The precision represents how much the content is tailored to the agent's taste. Note that a precise signal accurately conveys the true state of the world to an agent by helping her clearly understand the candidate's policies, track record, and character, which enables her to correctly infer if the candidate will win. A precise signal does not necessarily mean that the agent backs the planner's candidate, though. For example, if the candidate loses in the agent's community as per the true state of the world, a highly precise signal would induce the agent to oppose him. However, an imprecise signal is more likely to induce the agent into backing him since it obfuscates the true state of the world from the agent (possibly by being vague or confusing), thus increasing the chance that she thinks he is winning. Thus, the biased planner may be incentivized to decrease an agent's signal precision if he thinks that his candidate is losing. To tailor the ad to an agent, the planner must research how the agent best understands any content, which incurs costs for the planner. Even rendering the signal to be really imprecise is costly, as it still requires tailoring to the specific agent, e.g., the planner needs to know that an agent best understands focused and brief messages to be able to decrease precision by confusing her with long-form verbose, detailed articles. The planner selects the precisions so as to maximize the expected number of backers minus the cost incurred in generating the precisions.

## B  Social Learning Dynamics

We first summarize the sequence of developments at step $i$:

1. The planner determines $q_i$ based on $\mathcal{H}_i$ and, $q_i$ is observed by all agents.

2. Agent $i$ receives private signal $s_i$ such that $\mathbb{P}\{s_i = \omega\} = q_i$.

3. Agent $i$ takes action $a_i$ based on the observable history and their private signal with the aim of maximizing their own utility.

4. All other agents (and the planner) observe $a_i$ and the public belief at the end of step $i$, $b_{i+1}$ is updated accordingly.

Note that from (1), $q_i$ is a function of $\mathcal{H}_i$. Thus, if a random variable is conditioned upon both $q_i$ and $\mathcal{H}_i$, the conditioning on $q_i$ is redundant. We will use this principle throughout.

## B.1 PRIVATE BELIEF UPDATE

We now describe how agent $i$ chooses action $a_i$. Let $\alpha_i := \frac{b_i}{1-b_i}$ be the public likelihood ratio based on observed actions. Agent $i$ forms a private belief about $\omega$, $\tilde{b}_i = \mathbb{P}\{\omega = G | \mathcal{H}_i, q_i, s_i\}$, by combining the current public belief with her private signal, $s_i$. We consider a private likelihood ratio, $\tilde{\alpha}_i := \frac{\tilde{b}_i}{1-\tilde{b}_i}$. We now describe how $\tilde{\alpha}_i$ and $\tilde{b}_i$ are computed.

1. If private signal is $s_i = G$, then

$$
\begin{aligned}
\tilde{\alpha}_i = \frac{\tilde{b}_i}{1-\tilde{b}_i} &= \frac{\mathbb{P}\{\omega = G | \mathcal{H}_i, q_i, s_i = G\}}{\mathbb{P}\{\omega = B | \mathcal{H}_i, q_i, s_i = G\}} \\
&= \frac{\mathbb{P}\{s_i = G | \omega = G, \mathcal{H}_i, q_i\} \, \mathbb{P}\{\omega = G | \mathcal{H}_i, q_i\}}{\mathbb{P}\{s_i = G | \omega = B, \mathcal{H}_i, q_i\} \, \mathbb{P}\{\omega = B | \mathcal{H}_i, q_i\}} \\
&= \frac{q_i}{1-q_i} \frac{b_i}{1-b_i} \tag{8} \\
&= \frac{q_i}{1-q_i} \alpha_i \tag{9}
\end{aligned}
$$

In obtaining Equation (8), we apply the assumption that, conditioned upon $\omega$, agents' signals are independent of one another and the history to assert that $\mathbb{P}\{s_i = G | \omega = G, \mathcal{H}_i\} = \mathbb{P}\{s_i = G | \omega = G\} = q_i$. The same is done when conditioning upon $\omega = B$. We additionally use the fact that $q_i$ is a function $\mathcal{H}_i$.

2. If private signal is $s_i = B$, following the logic above,

$$
\tilde{\alpha}_i = \frac{1-q_i}{q_i} \alpha_i \tag{10}
$$

We let $y(b, q)$ and $z(b, q)$ denote the probability of realizing signal $G$ or $B$, with public belief $b$ and signal precision $q$ conditioned upon the history. By conditioning on $\omega$ and applying the Law of Total Probability, these probabilities can be expressed as follows:

$$
\begin{aligned}
y(b_i, q_i) &= \mathbb{P}\{s_i = G | \mathcal{H}_i\} \\
&= \sum_{k \in \{G, B\}} \mathbb{P}\{s_i = G | \mathcal{H}_i, \omega = k\} \, \mathbb{P}\{\omega = k | \mathcal{H}_i\} \\
&= q_i b_i + (1-q_i)(1-b_i) = 1 + 2 q_i b_i - q_i - b_i \tag{11}
\end{aligned}
$$

Similarly,

$$
z(b_i, q_i) = q_i + b_i - 2 q_i b_i \tag{12}
$$

Combining Equation (9), Equation (10), Equation (11), and Equation (12) yields the following:

$$
\tilde{b}_i = \mathbb{P}(\omega = G | \mathcal{H}_i, q_i, s_i) = f(b_i, q_i, s_i) = \begin{cases} \frac{q_i}{y(b_i, q_i)} b_i & s_i = G \\ \frac{1-q_i}{z(b_i, q_i)} b_i & s_i = B \end{cases} \tag{13}
$$

Note that if $s_i = G$ then $y(b_i, q_i) = \mathbb{P}(s_i = G | \mathcal{H}_i)$ cannot be 0. Similarly, if $s_i = B$, then $z(b_i, q_i)$ cannot be 0. Thus, the equation above is well-defined.

## B.2 AGENT ACTIONS AND PUBLIC BELIEF

Agent $i$'s action $a_i$ will be $G$ if $\tilde{b}_i$ exceeds 0.5 (equivalently, if $\tilde{\alpha}_i > 1$), $B$ if $\tilde{b}_i < 0.5$ (equivalently, if $\tilde{\alpha}_i < 1$), and $s_i$ otherwise as per the tie-breaking rule. This action and $i$'s precision $q_i$ are observed by all peers, resulting in an updated public belief $b_i$.

When $b_i < 1 - q_i$, then $\alpha_i < \frac{1-q_i}{q_i}$. Therefore, if $s_i = G$, $\tilde{\alpha}_i < 1$ by Equation (9); if $s_i = B$, $\tilde{\alpha}_i < \frac{(1-q_i)^2}{q_i^2} \leq 1$ (since $q_i \geq 0.5$) by Equation (10). Thus, no matter the private signal realization $s_i$, $a_i = B$ when $b_i < 1 - q_i$. Similar arguments can be used to show that when $b_i > q_i$, $a_i = G$. Intuitively these cases are when the strength of the public belief overpowers the precision of the private signal.

Now consider the case that $1 - q_i \leq b_i \leq q_i$. Then if $s_i = G$, $\tilde{\alpha}_i \geq 1$ from Equation (8) and since $b_i \geq 1 - q_i$. If $s_i = B$, then $\tilde{\alpha}_i \leq 1$ from Equation (10) and since $b_i \leq q_i$. Together with the tie-breaking rule, this results

in action $a_i = s_i$. This leads to Equation (2), reproduced below.

$$a_i = \begin{cases} s_i & 1 - q_i \leq b_i \leq q_i \\ G & q_i < b_i \\ B & q_i < 1 - b_i \end{cases}$$

Thus, agent $i$'s action is only informative in the first case of Equation (2) where it allows future agents to perfectly infer $i$'s private signal. Thus, in this case, public belief will update just as the private belief from Equation (13). In the latter two cases of Equation (2), the action contains no information about the true state. Thus, the public belief will not change. This yields Equation (3) reproduced below:

$$b_{i+1} = f(b_i, q_i) = \begin{cases} \tilde{b}_i & 1 - q_i \leq b_i \leq q_i \\ b_i & \text{o.w.} \end{cases} \tag{14}$$

### B.3 Markovianity and Sufficiency of Public Belief

We now prove that $\mathbb{P}\{a_i = a | \mathcal{H}_i, q_i\}$ is a function of $b_i, q_i$ for $a \in \{G, B\}$, regardless of the rest of $\mathcal{H}_i$. It follows then that

$$\mathbb{P}(a_i = a | \mathcal{H}_i, q_i) = \mathbb{P}(a_i = a | b_i, q_i) \tag{15}$$

From Equation (2), for the second and third cases, it is clear that $\mathbb{P}\{a_i = a | \mathcal{H}_i, q_i\}$ is a function of $b_i, q_i$ for $a \in \{G, B\}$, regardless of the rest of $\mathcal{H}_i$. We consider the first case next.

$$
\begin{aligned}
&\mathbb{P}(a_i = G | \mathcal{H}_i, q_i) \\
&= \sum_{w \in \{G, B\}} \mathbb{P}(\omega = w | \mathcal{H}_i, q_i) \, \mathbb{P}(a_i = G | \mathcal{H}_i, q_i, \omega = w) \\
&= b_i \, \mathbb{P}(a_i = G | \mathcal{H}_i, q_i, \omega = G) \\
&\quad + (1 - b_i) \, \mathbb{P}(a_i = G | \mathcal{H}_i, q_i, \omega = B) \\
&= b_i \, \mathbb{P}(s_i = G | b_i, q_i, \omega = G) \\
&\quad + (1 - b_i) \, \mathbb{P}(s_i = G | b_i, q_i, \omega = B) \\
&= b_i q_i + (1 - b_i)(1 - q_i)
\end{aligned} \tag{16}
$$

The first step applies the law of total probability. In the second step, using the fact that $q_i$ is a function of $\mathcal{H}_i$, we substitute $b_i$ for $\mathbb{P}(\omega = w | \mathcal{H}_i, q_i)$. In the third step, we use the fact that, for $1 - q_i \leq b_i \leq q_i$, $a_i = s_i$, and, further, that $s_i$ is independent of the history when conditioned upon $\omega$. The result follows.

This leads to the following theorem:

**Theorem 6.** *Markovianity of the Public Belief Process*

$$b_{i+1} = \mathbb{P}\{\omega = G | b_i, q_i, a_i\} \tag{17}$$

In other words, $b_i$ captures the entire information pertinent to the update of $b_{i+1}$ that is contained in $H_i$.

*Proof.* Note that, from step (1) in Appendix B, $q_i$ is a function of $\mathcal{H}_i$. It suffices to show

$$b_{i+1} = \frac{\mathbb{P}\{a_i \cap \omega = G | b_i, q_i\}}{\mathbb{P}\{a_i | b_i, q_i\}}$$

We will prove the claim via induction on $i$.

For our base case, consider $i = 1$ so that $\mathcal{H}_2 = \{b_1, (q_1, a_1)\}$ and $b_i = b_1$. An application of Bayes' Theorem provides the following:

$$
\begin{aligned}
b_2 &= \frac{\mathbb{P}\{\omega = G | b_1, q_1\} \, \mathbb{P}\{a_1 | \omega = G, b_1, q_1\}}{\mathbb{P}\{a_1 | b_1, q_1\}} = \frac{b_1 \, \mathbb{P}\{a_1 | \omega = G, b_1, q_1\}}{\mathbb{P}\{a_1 | b_1, q_1\}} \\
&= \frac{b_1 \, \mathbb{P}\{a_1 \cap \omega = G | b_1, q_1\}}{\mathbb{P}\{a_1 | b_1, q_1\} \, \mathbb{P}\{\omega = G | b_1, q_1\}} \\
&= \frac{\mathbb{P}\{a_1 \cap \omega = G | b_1, q_1\}}{\mathbb{P}\{a_1 | b_1, q_1\}}
\end{aligned}
$$

where the final step follows as the unconditioned belief of $\omega$ is equal to the prior belief $b_1$.

Now, assume the claim holds for all $i \leq n$.

$$b_{n+1} = \mathbb{P}\{\omega = G | \mathcal{H}_n, a_n, q_n\} = \frac{\mathbb{P}\{\omega = G | \mathcal{H}_n, q_n\} \, \mathbb{P}\{a_n | \omega = G, \mathcal{H}_n, q_n\}}{\mathbb{P}\{a_n | \mathcal{H}_n, q_n\}}$$

$$= \frac{b_n \, \mathbb{P}\{a_n | \omega = G, \mathcal{H}_n, q_n\}}{\mathbb{P}\{a_n | \mathcal{H}_n, q_n\}}$$

$$= \frac{b_n \, \mathbb{P}\{a_n \cap \omega = G | \mathcal{H}_n, q_n\}}{\mathbb{P}\{a_n | b_n, q_n\} \, \mathbb{P}\{\omega = G | \mathcal{H}_n, q_n\}}$$

$$= \frac{\mathbb{P}\{a_n \cap \omega = G | b_n, q_n\}}{\mathbb{P}\{a_n | b_n, q_n\}}$$

where the final two steps follow from Equation (15) and the definition of $b_n$.

This completes the proof. $\square$

Note that we have shown that the public belief is Markovian adapting the arguments in Banerjee (1992) and Bikhchandani et al. (1992). The modifications were required because of the differences in selections of the agents' precisions caused by the introduction of the central planner.

## C PROOFS

### C.1 DERIVATION OF EQUATION (4)

**Lemma 7.**

$$\mathbb{P}(a_i \neq \omega | b_i, q_i) = \min(b_i, 1 - b_i, 1 - q_i)$$

*Proof.* We prove this for each case of $a_i$ delineated in Equation (2). In the second case, we can reason as follows:

$$\mathbb{P}(a_i \neq \omega | b_i, q_i) = \mathbb{P}(\omega \neq G | b_i, q_i) = 1 - b_i$$

Similarly, in the third case of Equation (2), we obtain:

$$\mathbb{P}(a_i \neq \omega | b_i, q_i) = \mathbb{P}(\omega \neq B | b_i, q_i) = b_i$$

For the case when $1 - q_i \leq b_i \leq q_i$ and $a_i = s_i$ from Equation (2), we can reason as follows:

$$\mathbb{P}(a_i \neq \omega | b_i, q_i) = \sum_{a \in \{G, B\}} \mathbb{P}(a_i = a | b_i, q_i) \, \mathbb{P}(\omega \neq a | b_i, q_i, a_i = a)$$

$$= \mathbb{P}(s_i = G | b_i, q_i) \, \mathbb{P}(\omega \neq G | b_i, q_i, s_i = G)$$
$$\quad + \mathbb{P}(s_i = B | b_i, q_i) \, \mathbb{P}(\omega \neq B | b_i, q_i, s_i = B) \qquad \text{Eqn 2} \qquad (18)$$

$$= y(b_i, q_i) \, \mathbb{P}(\omega = B | b_i, q_i, s_i = G)$$
$$\quad + z(b_i, q_i) \, \mathbb{P}(\omega = G | b_i, q_i, s_i = B) \qquad \text{Eqns 11, 12, Thm 6} \qquad (19)$$

$$= \begin{cases} y(0, q_i) & b_i = 0 \\ y(b_i, q_i)(1 - \frac{b_i q_i}{y(b_i, q_i)}) + z(b_i, q_i) \frac{b_i(1 - q_i)}{z(b_i, q_i)} & b_i \neq 0 \end{cases} \qquad \text{Eqn 13} \qquad (20)$$

$$= \begin{cases} 1 - q_i & b_i = 0 \\ y(b_i, q_i) - b_i q_i + b_i(1 - q_i) & b_i \neq 0 \end{cases}$$

$$= \begin{cases} 1 - q_i & b_i = 0 \\ y(b_i, q_i) - b_i q_i + b_i - b_i q_i & b_i \neq 0 \end{cases}$$

$$= \begin{cases} 1 - q_i & b_i = 0 \\ 1 - q_i & b_i \neq 0 \end{cases}$$

Combining the three cases yields the claim. $\square$

## C.2    PROOF OF THEOREM 1: MYOPIC ALTRUISTIC POLICY

Applying Equation (4) to $r_A(b, q)$ in Equation (5) allows us to reason as follows

$$\frac{\partial r_A(b, q)}{\partial q} = \begin{cases} -\dot{\beta}(q) & q < \max(b, 1 - b) \\ -\dot{\beta}(q) + C & q \geq \max(b, 1 - b) \end{cases} \tag{21}$$

As stated in Section 3.2, we can restrict the altruistic planner to precision $q \in [p, 1]$ because social welfare is increasing in precision (see Appendix C.9). Accordingly, $\beta(p) = 0$.

Because $\beta(\cdot)$ is increasing, the first case of Equation (21) leads to optimal precision $p$ if $p < \max(b, 1 - b)$. That is, if the signal is not precise enough to influence the agent's action, then the planner will not expend resources for it. In the latter case, because $\beta(\cdot)$ is increasing and concave, $\dot{\beta}(\cdot)$ is positive and decreasing. Thus, $r_A(b, q)$, if the optimal lies in this range it must be attained at $q = 1$.

We now identify the regime in which the latter case is optimal.

$$r_A(b, p) < r_A(b, 1)$$
$$-C \min(b, 1 - b, 1 - p) < -\beta(1)$$
$$\min(b, 1 - b, 1 - p) > \frac{\beta(1)}{C}$$

If $1 - p \leq \frac{\beta(1)}{C}$, the above inequality is never satisfied, and it is never optimal to select precision 1. Otherwise, the inequality above yields $b \in \left[\frac{\beta(1)}{C}, 1 - \frac{\beta(1)}{C}\right]$. This yields Theorem 1. ∎

## C.3    PROOF OF THEOREM 2: ALTRUISTIC VALUE FUNCTION CONVEXITY

**Proof Sketch**    We inductively prove that the expected $k$-th stage reward, i.e., the expected utility of the planner from the control and action of the $k$-th agent, is convex. The instantaneous reward (Equation (5)), which is convex with respect to public belief, provides our base case.

The first challenge encountered is the unusual nature of the public belief update. Although the state space is uncountably infinite, the belief update only takes support on a maximum of 2 values. To manage this, we define a decision tree, i.e., the complete binary tree of all possible trajectories once an initial belief and policy are fixed. Each node has two children corresponding to each possible signal realization the next agent might receive. The root is the expected instantaneous reward at time 1, i.e., from the first agent's action. The induction moves down the levels of this tree with the $k$-th level containing $2^{k-1}$ nodes, each associated with a sequence of realizations of $k - 1$ signals.

We then show that for a node in the $(k - 1)$-th level that has convex expected reward, its two children in the $k$-th level satisfy the same property. This is where we must deal with the dependence of agents' actions on public belief. Note that even when applying the same precision and receiving the same signal realization, two agents beginning at different public beliefs may take opposing actions (see Equation (2)). Thus, standard results that provide easy ways of bounding the future terms of the Markov process do not apply. Here, our specific belief update is actually helpful. We can leverage the fact that Bayesian updates are martingales (i.e., $\mathbb{E}[b_{i+1}] = b_i$). Along with the convexity of instantaneous rewards, this allows us to complete the inductive step and, subsequently, the proof.

**Preliminaries**    We first prove the following lemma containing a few useful properties of the optimal value function $V_A^*(\cdot)$.

**Lemma 8.**
$V_A^*(\cdot)$ is non-positive, $V_A^*(b) = V_A^*(1 - b) \forall b \in [0, 1]$, and $V_A^*(0) = V_A^*(1) = 0$ with $\pi_A^*(0) = \pi_A^*(1) = p$.

*Proof.*
*Non-positivity*
$r(b_i, q_i) \leq 0 \quad \forall b_i, q_i \rightarrow V_A^* \leq 0$

*Symmetry*
Let $n_i = 1 - b_i$ and let $\Lambda_n$ and $\Lambda_b$ be the decision trees rooted at states $n_0$ and $b_0$, respectively. Showing that $\Lambda_n$ and $\Lambda_b$ are symmetric suffices to show the symmetry of $V^*(\cdot)$. This will be accomplished by showing that the expected rewards, belief state values, and branching probabilities of the trees are reflections of one another.

First consider the instantaneous reward function $r_A$. Note that the decision function of the agents when choosing their actions is symmetric from Equation (2), thus, $\mathbb{P}\{a_i = G | b_i\} = \mathbb{P}\{a_i = B | n_i\}$. Applying this, we

obtain

$$
\begin{aligned}
r(n_i, q_i) &= -\beta(q_i) - C n_i \, \mathbb{P}\{a_i = B | \omega = G, n_i\} - C(1 - n_i) \, \mathbb{P}\{a_i = G | \omega = B, n_i\} \\
&= -\beta(q_i) - C(1 - b_i) \, \mathbb{P}\{a_i = B | \omega = G, n_i\} - C b_i \, \mathbb{P}\{a_i = G | \omega = B, n_i\} \\
&= -\beta(q_i) - C(1 - b_i) \, \mathbb{P}\{a_i = G | \omega = B, b_i\} - C b_i \, \mathbb{P}\{a_i = B | \omega = G, b_i\} \\
&= r(b_i, q_i)
\end{aligned}
$$

Thus, the instantaneous reward function is symmetric about 0.5.

Now consider the belief state update when beginning with belief $n_i$ assuming signal precision $q_i$. we will use $\bar{x}$ and $\underline{x}$ to represent updated beliefs from $x$ after observing action $G$ or $B$, respectively. From Equation (13):

$$
\begin{aligned}
\bar{n}_i &= \frac{q_i}{y(n_i, q_i)} n_i = \frac{q_i}{z(b_i, q_i)}(1 - b_i) = 1 - \frac{1 - q_i}{z(b_i, q_i)} b_i = 1 - \underline{b}_i \\
\underline{n}_i &= \frac{1 - q_i}{z(n_i, q_i)} n_i = \frac{q_i}{y(b_i, q_i)}(1 - b_i) = 1 - \frac{q_i}{y(b_i, q_i)} b_i = 1 - \bar{b}_i
\end{aligned}
$$

Thus, the belief updates are also symmetric. Combining this with the symmetry of the reward we obtain:

$$
\begin{aligned}
r(\bar{n}_i) &= r(1 - \underline{b}_i) = r(\underline{b}_i) \\
r(\underline{n}_i) &= r(1 - \bar{b}_i) = r(\bar{b}_i)
\end{aligned}
$$

Thus, the expected rewards and belief state values of $\Lambda_n$ and $\Lambda_b$ are mirror images of one another.

Finally, consider the branching probabilities which we will denote as $y(b, q) = 1 + 2bq - b - q$ for observing action $G$ and $z(b, q) = 1 - y(b, q)$ for observing action $B$.

$$
\begin{aligned}
y(n, q) &= 1 + 2nq - n - q = 1 + 2(1 - b)q - (1 - b) - q = b + q - 2bq = z(b, q) \\
z(n, q) &= 1 - y(n, q) = 1 - z(b, q) = y(b, q)
\end{aligned}
$$

Thus, the branching probabilities of $\Lambda_n$ and $\Lambda_b$ are also symmetric, completing the proof.

*Extremal values*
We will show that $V_A^*(0) = 0$ and rely on the symmetry about 0.5 proven above to show that $V_0^*(1) = 0$.

From Equation (3), when $b_0 = 0$, $b_0^+ = b_0^- = 0$ for any value of precision. Thus, regardless of signal realizations or control actions, the agents' actions and, subsequently, the public belief, will remain unchanged. That is, $b_i = 0 \forall i$ and, from Equation (2), $a_i = B \forall i$.

From Equation (5), we can then write:

$$
\begin{aligned}
V_A^*(0) &= \max_{\pi \in \Pi} \mathbb{E}\left[ \sum_{i=0}^{\infty} \delta^i r(b_i, \pi(b_i)) \right] \\
&= \max_{\pi \in \Pi} \mathbb{E}\left[ \sum_{i=0}^{\infty} \delta^i r(0, \pi(0)) \right] \\
&= \max_{\pi \in \Pi} \mathbb{E}\left[ \sum_{i=0}^{\infty} \delta^i \left[ -\beta(\pi(0)) - C b_i \, \mathbb{P}\{a_i = B | \omega = G, b_i = 0\} - C(1 - b_i) \, \mathbb{P}\{a_i = G | \omega = B, b_i = 0\} \right] \right] \\
&= \max_{\pi \in \Pi} \mathbb{E}\left[ \sum_{i=0}^{\infty} \delta^i \left[ -\beta(q_i) \right] \right] \tag{22}
\end{aligned}
$$

Thus, to attain $V_A^*(0)$, it suffices to choose $\pi(0)$ such that $\beta(\pi(0))$ is minimized for each $i$. Recall that $\beta(\cdot)$ is increasing on $[0.5, 1]$ and takes its minimum at $\beta(p) = 0$. Thus, by choosing $q_i = p \forall i$, Equation 22 yields $V_A^*(0) = 0$ with corresponding optimal policy $\pi_A^*(0) = p$.

The aforementioned symmetry about 0.5, shows that $V_A^*(1) = 0$, completing the proof. $\qquad \square$

We now introduce the concept of *decision trees* for Markovian and more general policies. A policy can be represented by a decision tree, which is a complete binary tree as in figure 3. Every node of the decision tree represents the state of the system, i.e., the corresponding belief value when the controller sends its private signal to the agent corresponding to the level of the node. The branches represent the actions taken by the agent corresponding to the level once it receives its private signal. The controller's policies (i.e. choice of precision) determine the probabilities of the actions from each node and the belief resulting from the action. The probabilities of an action at a certain node can be considered the weight of the branch. A decision tree is uniquely

identified by the belief values associated with nodes and the weights of the branches. For a deterministic stationary Markovian policy, the controller's choice of precision, and therefore the probability associated with an action, is a deterministic function of the belief value of the node, regardless of the level of the node in the tree and the path to the node from the root. Thus, the decision tree of any such policy, i.e. the belief values at the nodes and the weights of the branches, is determined entirely by the value of the initial state. For an arbitrary policy of the controller, the controller's decision at a given node can depend on the entire path leading to the node.

**Proof of Theorem 2**    We now prove the convexity of $V_A^*(\cdot)$ with respect to public belief.

*Proof.*

We will first show that, for fixed precision, the instantaneous reward $r_A$ is convex in public belief. We can rewrite $r_A$ as follows using Lemma 7:

$$r_A(x, q) = -\beta(q) - C\min(x, 1 - x, 1 - q) \tag{23}$$

The pointwise minimum of linear functions is concave. As we are subtracting the minimum, the overall function is convex with respect to $x$.

We consider arbitrary $x_0, m_0, n_0$ such that $tm_0 + (1 - t)n_0 = x_0$ for some $t \in [0, 1]$. To show the convexity of $V_A^*(\cdot)$ we must show that $V_A^*(x_0) \leq tV_A^*(m_0) + (1 - t)V_A^*(n_0)$. By definition of the optimal value function, $V_A^*(x) \geq V_A^\pi(x)$ for any policy $\pi$. Thus, it will suffice to show that there exist $\tilde{\pi}$ such that

$$V_A^*(x_0) \leq tV_A^{\tilde{\pi}}(m_0) + (1 - t)V_A^{\tilde{\pi}}(n_0) \tag{24}$$

We refer to the decision tree of $\pi^*$ when the initial state is $x_0$ as $\Lambda$.

We consider $\tilde{\pi}$ to be a policy of the controller that chooses the precision at each node to be the same as what $\Lambda$ does at the counterpart node. The nodes of decision trees are considered to be counterparts if they can be reached via the same sequence of actions from the root (see figure 3 for illustration). Thus, the precision chosen by this policy is the same as what $\Lambda$ does after the same set of actions of the agents. We refer to the decision trees for $\tilde{\pi}$ with initial state $m_0$ (respectively, $n_0$) as $\Lambda_m$ ($\Lambda_n$).

We now illustrate the policy $\tilde{\pi}$ for initial values $m_0$ and $n_0$ considering the decision trees $\Lambda_m$ and $\Lambda_n$. While doing so, we illustrate counterpart nodes and the weights of the branches. We will select one path of actions $\lambda$ out of these decision trees by specifying set $\mathcal{G}$ as the epochs where action $G$ is taken and $\mathcal{B}$ as its complement. Refer to figure 3 for illustration. Considering the decision tree $\Lambda$, we will refer to $x_i$ as the state at epoch $i$ when beginning with belief $x_0$ and following $\pi^*$ along this path of actions. That is, $x_i$ assumes one of $2^i$ possible belief values at epoch $i$ starting from $x_0$. In decision tree $\Lambda_m$ (respectively, $\Lambda_n$), we use $m_i$ ($n_i$) to denote the public belief after beginning with $m_0$ ($n_0$), following the path $\lambda$. Node $m_i$ (respectively $n_i$) in decision tree $\Lambda_m$ ($\Lambda_n$) is the counterpart of node $x_i$ from decision tree $\Lambda$. Note that, $\tilde{\pi}(m_i)$ can now be illustrated notationally. Specifically, $\tilde{\pi}(m_i) = \tilde{\pi}(n_i) = \pi_A^*(x_i)$. Similarly, $\tilde{\pi}$ can be defined at each node of $\Lambda_m$, considering the action taken by $\pi^*$ at each counterpart node in $\Lambda$. Note that $y(x_i, \pi_A^*(x_i))$ (respectively, $z(x_i, \pi_A^*(x_i))$) is the weight of the branch corresponding to $G$ ($B$) emanating from $x_i$ in $\Lambda$. Because $\tilde{\pi}(m_i) = \tilde{\pi}(n_i) = \pi_A^*(x_i)$, $y(m_i, \tilde{\pi}(m_i)) = y(m_i, \pi_A^*(x_i))$ is the weight of the branch emanating from $m_i$ for action $G$ in $\Lambda_m$. This extends similarly to branches associated with action $B$ and $\Lambda_n$. Thus, the three decision trees differ in both the values of the nodes and the weights of the branches. Since $\pi^*$ is deterministic, so is $\tilde{\pi}$. Therefore, from Equation (13) and Equation (3), $x_i$, $m_i$, and $n_i$ are deterministic quantities once the specific path of actions is specified by $\mathcal{G}$.

Now let $\phi_k^\pi(b_0) = \mathbb{E}[r_A(b_k, \pi(b_k)]$ be the expected instantaneous reward at epoch $k$ when beginning with state $b_0$ and applying policy $\pi$ and $\phi^*(x) = \phi^{\pi_A^*}(x)$. Applying Fubini's Theorem, we can rewrite our value function as

$$V_A^\pi(b_0) = \mathbb{E}\left[\sum_{i=0}^{\infty}\delta^i r_A(b_i, \pi(b_i))\right] = \sum_{i=0}^{\infty}\delta^i\mathbb{E}[r_A(b_i, \pi(b_i))] = \sum_{i=0}^{\infty}\delta^i\phi_i^\pi(b_0) \tag{25}$$

Thus, we can rewrite (24), as

$$\sum_{i=0}^{\infty}\delta^i\phi_i^*(x_0) \leq t\left[\sum_{i=0}^{\infty}\delta^i\phi_i^{\tilde{\pi}}(m_0)\right] + (1 - t)\left[\sum_{i=0}^{\infty}\delta^i\phi_i^{\tilde{\pi}}(n_0)\right] \tag{26}$$

To prove this, we show that for all $i = 0, 1, 2, \ldots$

$$\phi_i^*(x_0) \leq t\phi_i^{\tilde{\pi}}(m_0) + (1 - t)\phi_i^{\tilde{\pi}}(n_0) \tag{27}$$

leading directly to (26), (24), and finally the convexity of $V^*(\cdot)$ with respect to public belief.

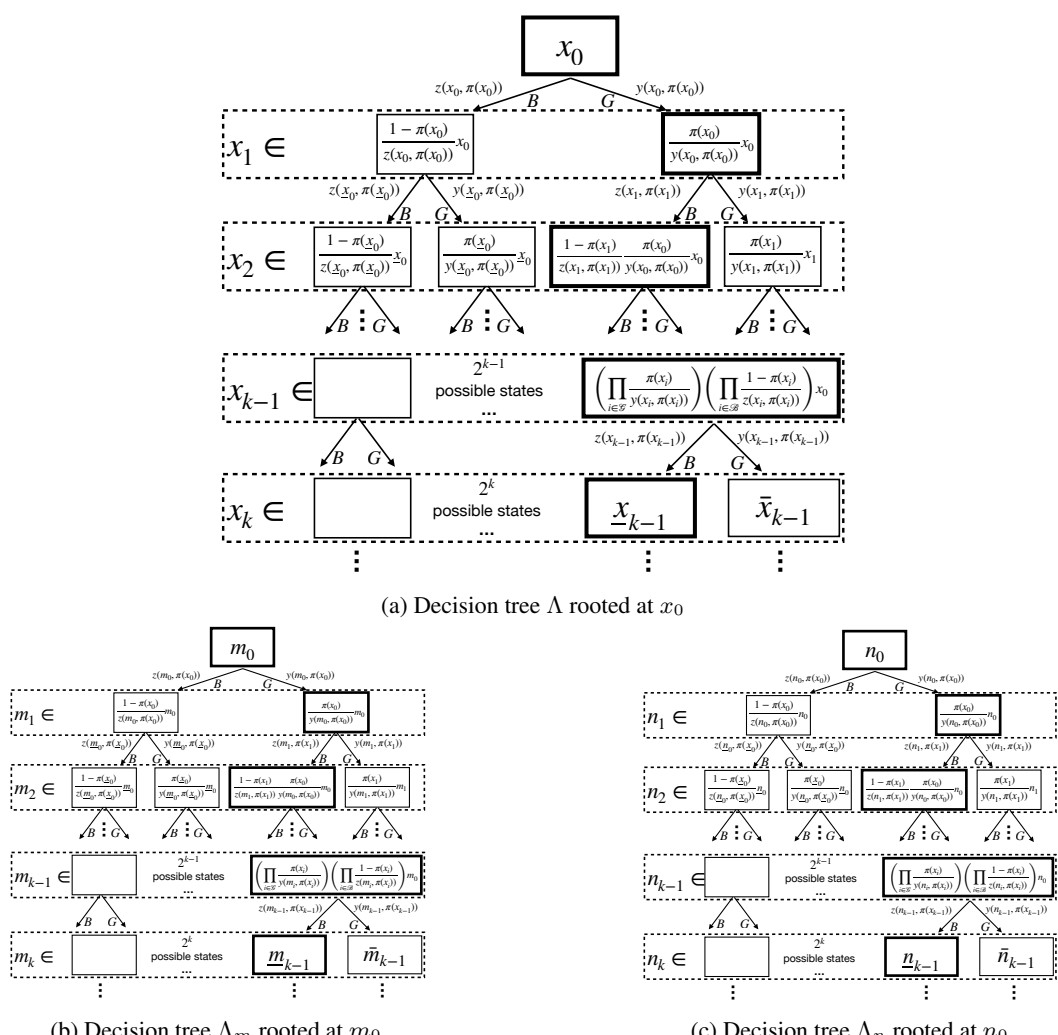

(a) Decision tree $\Lambda$ rooted at $x_0$

(b) Decision tree $\Lambda_m$ rooted at $m_0$

(c) Decision tree $\Lambda_n$ rooted at $n_0$

Figure 3: Here we depict the belief evolution in the form of three binary trees. The tree begins with state $x_0, m_0$, and $n_0$, respectively, and level $i$ contains $2^i$ possible states reachable after $i$ epochs. Left branches correspond to action $B$ leading to updated belief $\underline{x}_0$ (respectively, $\underline{m}_0$ and $\underline{n}_0$). Right branches correspond to action $G$ leading to updated belief $\bar{x}_0$ (respectively, $\bar{m}_0$ and $\bar{n}_0$). Each branch is labeled with the corresponding probability, and the bold states indicate an example path, like $\lambda$. In the example path, $0 \in \mathcal{G}$ and $1, k-1 \in \mathcal{B}$. The bold states in each level of the three trees are counterparts.

Now let us consider general $i = k$. $\phi_{k-1}^\pi(x_0)$ will have $2^{k-1}$ terms each corresponding to one path of possible actions in the set $\{G, B\}^{k-2}$. Thus, at epoch $k-1$, the path $\lambda$ will yield the following term in $\phi_{k-1}^*(x_0)$

$$\left(\prod_{i \in \mathcal{G}} y(x_i, \pi_A^*(x_i))\right) \left(\prod_{i \in \mathcal{B}} z(x_i, \pi_A^*(x_i))\right) r_A(x_{k-1}, \pi_A^*(x_{k-1})) \tag{28}$$

We now define $P_\lambda^*(x_0) = \left(\prod_{i \in \mathcal{G}} y(x_i, \pi_A^*(x_i))\right) \left(\prod_{i \in \mathcal{B}} z(x_i, \pi_A^*(x_i))\right)$, that is, the probability of following the specified path $\lambda$ when beginning with state $x_0$ and following policy $\pi_A^*$. Again, we will use $\bar{x}_i$ and $\underline{x}_i$ to represent updated beliefs from $x_i$ after observing action $G$ or $B$, respectively. Thus, $\bar{x}_i$ and $\underline{x}_i$ constitute belief values of the children nodes (in the $(i+1)$th level) of the node with belief value $x_i$ in the $i$-th level. Refer to levels $k-1$ and $k$ in figure 3a. Considering the children $\bar{x}_{k-1}$ and $\underline{x}_{k-1}$ (in the $k$-th level) of the node $x_{k-1}$ (in the $(k-1)$-th level), we obtain the following two terms in $\phi_k^*(x_0)$:

$$P_\lambda^*(x_0) \left[ y(x_{k-1}, \pi_A^*(x_{k-1})) r_A(\bar{x}_{k-1}, \pi_A^*(\bar{x}_{k-1})) + z(x_{k-1}, \pi_A^*(x_{k-1})) r_A(\underline{x}_{k-1}, \pi_A^*(\underline{x}_{k-1})) \right] \tag{29}$$

Similarly, we define $P_\lambda^{\tilde\pi}(m_0) = \left(\prod_{i \in \mathcal{G}} y(m_i, \tilde\pi(m_i))\right) \left(\prod_{i \in \mathcal{B}} z(m_i, \tilde\pi(m_i))\right)$, that is, the probability of following the specified path $\lambda$ when beginning with state $m_0$ and following policy $\tilde\pi$. Because $\tilde\pi(m_i) = \pi_A^*(x_i)$,

$P_\lambda^{\tilde{\pi}}(m_0) = \left(\prod_{i \in \mathcal{G}} y(m_i, \pi_A^*(x_i))\right)\left(\prod_{i \in \mathcal{B}} z(m_i, \pi_A^*(x_i))\right)$. We also define $P_\lambda^{\tilde{\pi}}(n_0)$ by replacing $\{m_i\}$ with $\{n_i\}$.

The corresponding terms in $\phi_k^{\tilde{\pi}}(m_0)$ and $\phi_k^{\tilde{\pi}}(n_0)$ are

$$P_\lambda^{\tilde{\pi}}(m_0)\left[y(m_{k-1}, \pi_A^*(x_{k-1}))r_A(\bar{m}_{k-1}, \pi_A^*(\bar{x}_{k-1})) + z(m_{k-1}, \pi_A^*(x_{k-1}))r_A(\underline{m}_{k-1}, \pi_A^*(\underline{x}_{k-1}))\right] \quad (30)$$

$$P_\lambda^{\tilde{\pi}}(n_0)\left[y(n_{k-1}, \pi_A^*(x_{k-1}))r_A(\bar{n}_{k-1}, \pi_A^*(\bar{x}_{k-1})) + z(n_{k-1}, \pi_A^*(x_{k-1}))r_A(\underline{n}_{k-1}, \pi_A^*(\underline{x}_{k-1}))\right] \quad (31)$$

We will show that

$$P_\lambda^*(x_0)\left[y(x_{k-1}, \pi_A^*(x_{k-1}))r_A(\bar{x}_{k-1}, \pi_A^*(\bar{x}_{k-1})) + z(x_{k-1}, \pi_A^*(x_{k-1}))r_A(\underline{x}_{k-1}, \pi_A^*(\underline{x}_{k-1}))\right]$$

$$\leq tP_\lambda^{\tilde{\pi}}(m_0)\left[y(m_{k-1}, \pi_A^*(x_{k-1}))r_A(\bar{m}_{k-1}, \pi_A^*(\bar{x}_{k-1})) + z(m_{k-1}, \pi_A^*(x_{k-1}))r_A(\underline{m}_{k-1}, \pi_A^*(\underline{x}_{k-1}))\right]$$

$$+ (1-t)P_\lambda^{\tilde{\pi}}(n_0)\left[y(n_{k-1}, \pi_A^*(x_{k-1}))r_A(\bar{n}_{k-1}, \pi_A^*(\bar{x}_{k-1})) + z(n_{k-1}, \pi_A^*(x_{k-1}))r_A(\underline{n}_{k-1}, \pi_A^*(\underline{x}_{k-1}))\right]$$
$$(32)$$

Similarly, each pair of children for each term in the $(k-1)$-th level of the belief tree can be combined and the corresponding inequalities can be obtained following the same process. Summing the resulting inequalities will yield (27), leading to (26), (24), and the convexity of $V^*(\cdot)$.

*Proof of (32)*

We now focus on the latter of the two terms of (29). We will show the following two properties

$$\underline{x}_{k-1} = t\left(\prod_{i \in \mathcal{G}} \frac{y(m_i, \pi_A^*(x_i))}{y(x_i, \pi_A^*(x_i))}\right)\left(\prod_{i \in \mathcal{B} \cup \{k-1\}} \frac{z(m_i, \pi_A^*(x_i))}{z(x_i, \pi_A^*(x_i))}\right)\underline{m}_{k-1}$$

$$+ (1-t)\left(\prod_{i \in \mathcal{G}} \frac{y(n_i, \pi_A^*(x_i))}{y(x_i, \pi_A^*(x_i))}\right)\left(\prod_{i \in \mathcal{B} \cup \{k-1\}} \frac{z(n_i, \pi_A^*(x_i))}{z(x_i, \pi_A^*(x_i))}\right)\underline{n}_{k-1} \quad (33)$$

$$1 = t\left(\prod_{i \in \mathcal{G}} \frac{y(m_i, \pi_A^*(x_i))}{y(x_i, \pi_A^*(x_i))}\right)\left(\prod_{i \in \mathcal{B} \cup \{k-1\}} \frac{z(m_i, \pi_A^*(x_i))}{z(x_i, \pi_A^*(x_i))}\right)$$

$$+ (1-t)\left(\prod_{i \in \mathcal{G}} \frac{y(n_i, \pi_A^*(x_i))}{y(x_i, \pi_A^*(x_i))}\right)\left(\prod_{i \in \mathcal{B} \cup \{k-1\}} \frac{z(n_i, \pi_A^*(x_i))}{z(x_i, \pi_A^*(x_i))}\right) \quad (34)$$

Defining $\tilde{t} = t\left(\prod_{i \in \mathcal{G}} \frac{y(m_i, \pi_A^*(x_i))}{y(x_i, \pi_A^*(x_i))}\right)\left(\prod_{i \in \mathcal{B} \cup \{k-1\}} \frac{z(m_i, \pi_A^*(x_i))}{z(x_i, \pi_A^*(x_i))}\right)$, it follows from (33) and (34) that

$$\underline{x}_{k-1} = \tilde{t}\underline{m}_{k-1} + (1-\tilde{t})\underline{n}_{k-1} \quad (35)$$

From the expressions for $\tilde{t}$, $P_\lambda^*(x_0)$, $P_\lambda^{\tilde{\pi}}(m_0)$, it follows that:

$$\tilde{t}z(x_{k-1}, \pi_A^*(x_{k-1}))P_\lambda^*(x_0) = tz(m_{k-1}, \pi_A^*(x_{k-1}))P_\lambda^{\tilde{\pi}}(m_0) \quad (36)$$

Similarly, we can show

$$\tilde{t}z(x_{k-1}, \pi_A^*(x_{k-1}))P_\lambda^*(x_0) = (1-t)z(n_{k-1}, \pi_A^*(x_{k-1}))P_\lambda^{\tilde{\pi}}(n_0) \quad (37)$$

We apply this as follows to the latter term of (29):

$$P_\lambda^*(x_0)z(x_{k-1}, \pi_A^*(x_{k-1}))r_A(\underline{x}_{k-1}, \pi_A^*(\underline{x}_{k-1}))$$

$$\leq P_\lambda^*(x_0)z(x_{k-1}, \pi_A^*(x_{k-1}))$$

$$\left[\tilde{t}r_A(\underline{m}_{k-1}, \pi_A^*(\underline{x}_{k-1})) + (1-\tilde{t})r_A(\underline{n}_{k-1}, \pi_A^*(\underline{x}_{k-1}))\right] \qquad \text{(Convexity of } r, \text{Eqn 35)}$$

$$= t\left[P_\lambda^{\tilde{\pi}}(m_0)z(m_{k-1}, \pi_A^*(x_{k-1}))r_A(\underline{m}_{k-1}, \pi_A^*(\underline{x}_{k-1}))\right]$$

$$+ (1-t)\left[P_\lambda^{\tilde{\pi}}(n_0)z(n_{k-1}, \pi_A^*(x_{k-1}))r_A(\underline{n}_{k-1}, \pi_A^*(\underline{x}_{k-1}))\right] \qquad \text{(Eqns 36, 37)} \quad (38)$$

Applying the same argument to the first term of (29) yields:

$$P_\lambda^*(x_0)y(x_{k-1}, \pi_A^*(x_{k-1}))r_A(\bar{x}_{k-1}, \pi_A^*(\bar{x}_{k-1}))$$

$$\leq t\left[P_\lambda^{\tilde{\pi}}(m_0)y(m_{k-1}, \pi_A^*(x_{k-1}))r_A(\bar{m}_{k-1}, \pi_A^*(\bar{x}_{k-1}))\right]$$

$$+ (1-t)\left[P_\lambda^{\tilde{\pi}}(n_0)y(n_{k-1}, \pi_A^*(x_{k-1}))r_A(\bar{n}_{k-1}, \pi_A^*(\bar{x}_{k-1}))\right] \quad (39)$$

Summing (38) and (39) yields (32).

We now complete the proof of (32) and, therefore, the entire proof, by proving (33) and (34).

*Proof of (33)*

From the recurrence relation in Equation (13) and Equation (3), we can write $\underline{x}_{k-1}, \underline{m}_{k-1}$, and $\underline{n}_{k-1}$ as the product of all belief updates with $x_0, m_0$, and $n_0$, respectively, as follows. Each term in the products corresponds to the multiplicative state update after a single action.

$$\underline{x}_{k-1} = \left( \prod_{i \in \mathcal{G}} \frac{\pi_A^*(x_i)}{y(x_i, \pi_A^*(x_i))} \right) \left( \prod_{i \in \mathcal{B} \cup \{k-1\}} \frac{1 - \pi_A^*(x_i)}{z(x_i, \pi_A^*(x_i))} \right) x_0 \tag{40}$$

$$\underline{m}_{k-1} = \left( \prod_{i \in \mathcal{G}} \frac{\pi_A^*(x_i)}{y(m_i, \pi_A^*(x_i))} \right) \left( \prod_{i \in \mathcal{B} \cup \{k-1\}} \frac{1 - \pi_A^*(x_i)}{z(m_i, \pi_A^*(x_i))} \right) m_0 \tag{41}$$

$$\underline{n}_{k-1} = \left( \prod_{i \in \mathcal{G}} \frac{\pi_A^*(x_i)}{y(n_i, \pi_A^*(x_i))} \right) \left( \prod_{i \in \mathcal{B} \cup \{k-1\}} \frac{1 - \pi_A^*(x_i)}{z(n_i, \pi_A^*(x_i))} \right) n_0 \tag{42}$$

These equations may be best understood via the illustration in figure 3. We then reorganize terms to relate $\underline{x}_{k-1}$ to $\underline{m}_{k-1}$ and $\underline{n}_{k-1}$, using the fact that $x_0 = tm_0 + (1 - t)n_0$.

$$\underline{x}_{k-1} = \left( \prod_{i \in \mathcal{G}} \frac{\pi_A^*(x_i)}{y(x_i, \pi_A^*(x_i))} \right) \left( \prod_{i \in \mathcal{B} \cup \{k-1\}} \frac{1 - \pi_A^*(x_i)}{z(x_i, \pi_A^*(x_i))} \right) [tm_0 + (1 - t)n_0] \qquad \text{(from 40)}$$

$$= t \left( \prod_{i \in \mathcal{G}} \frac{y(m_i, \pi_A^*(x_i))}{y(x_i, \pi_A^*(x_i))} \right) \left( \prod_{i \in \mathcal{B} \cup \{k-1\}} \frac{z(m_i, \pi_A^*(x_i))}{z(x_i, \pi_A^*(x_i))} \right) \underline{m}_{k-1}$$

$$+ (1 - t) \left( \prod_{i \in \mathcal{G}} \frac{y(n_i, \pi_A^*(x_i))}{y(x_i, \pi_A^*(x_i))} \right) \left( \prod_{i \in \mathcal{B} \cup \{k-1\}} \frac{z(n_i, \pi_A^*(x_i))}{z(x_i, \pi_A^*(x_i))} \right) \underline{n}_{k-1} \qquad \text{(from 41 and 42)} \tag{43}$$

This proves (33).

*Proof of (34)*

We first consider the numerator of the first term in (34): $\prod_{i \in \mathcal{G}} y(m_i, \pi_A^*(x_i)) \prod_{i \in \mathcal{B} \cup \{k-1\}} z(m_i, \pi_A^*(x_i))$. We now prove that this can be written as $\alpha_0 m_0 + \beta_0$ for some $\alpha_0, \beta_0$ which depend only on $\{\pi_A^*(x_i)\}_{i=1,\dots,k-1}$ and the nature of the dependence is determined by the sequence of actions along this path in stages 1 to $k - 1$. We will prove this via induction working backwards from the final term (i.e. the $(k-1)$-th term) of the product. Therefore our base case will be the the final term of this product which can be either $y(m_{k-1}, \pi_A^*(x_{k-1}))$ or $z(m_{k-1}, \pi_A^*(x_{k-1}))$, depending on the preceding action (in the $(k-1)$-th stage).

$$y(m_{k-1}, \pi_A^*(x_{k-1})) = 1 + 2m_{k-1}\pi_A^*(x_{k-1}) - m_{k-1} - \pi_A^*(x_{k-1})$$
$$= (2\pi_A^*(x_{k-1}) - 1)m_{k-1} + (1 - \pi_A^*(x_{k-1}))$$
$$z(m_{k-1}, \pi_A^*(x_{k-1})) = m_{k-1} + \pi_A^*(x_{k-1}) - 2m_{k-1}\pi_A^*(x_{k-1})$$
$$= (1 - 2\pi_A^*(x_{k-1}))m_{k-1} + (\pi_A^*(x_{k-1}))$$

Thus, the final term of the product is of the form $\alpha_{k-1}m_{k-1} + \beta_{k-1}$ where $\alpha_{k-1}$ and $\beta_{k-1}$ depend only upon $\pi_A^*(x_{k-1})$, and the nature of the dependence is determined entirely by the $(k-1)$-th action (since this action determines whether the final term is $y(\cdot)$ or $z(\cdot)$).

For the inductive step, assume that $\prod_{i \in \mathcal{G}, i>j} y(m_i, \pi_A^*(x_i)) \prod_{i \in \mathcal{B} \cup \{k-1\}, i>j} z(m_i, \pi_A^*(x_i))$ can be written as $\alpha_{j+1}m_{j+1} + \beta_{j+1}$ for some $\alpha_{j+1}, \beta_{j+1}$ which depend only $\{\pi_A^*(x_i)\}_{i>j}$ and the nature of the dependence is determined entirely by the sequence of actions in this path for $i > j$ to $i = k - 1$. We will now show that $\prod_{i \in \mathcal{G}, i>j-1} y(m_i, \pi_A^*(x_i)) \prod_{i \in \mathcal{B} \cup \{k-1\}, i>j-1} z(m_i, \pi_A^*(x_i))$ can be written as $\alpha_j m_j + \beta_j$ again with $\alpha_j, \beta_j$ depending only on $\{\pi_A^*(x_i)\}_{i>j-1}$ with the nature of the dependence determined by the sequence of actions on this path for $i > j - 1$ to $i = k - 1$. We define $f(b, q, a) = \begin{cases} y(b, q) & a = G \\ z(b, q) & a = B \end{cases}$ and $g(q, a) = \begin{cases} q & a = G \\ 1 - q & a = B \end{cases}$.

Thus,

$$\prod_{i \in \mathcal{G}, i > j-1} y(m_i, \pi_A^*(x_i)) \prod_{i \in \mathcal{B} \cup \{k-1\}, i > j-1} z(m_i, \pi_A^*(x_i))$$

$$= f(m_j, \pi_A^*(x_j), a_{j+1}) \prod_{i \in \mathcal{G}, i > j} y(m_i, \pi_A^*(x_i)) \prod_{i \in \mathcal{B} \cup \{k-1\}, i > j} z(m_i, \pi_A^*(x_i))$$

$$= f(m_j, \pi_A^*(x_j), a_{j+1}) [\alpha_{j+1} m_{j+1} + \beta_{j+1}] \qquad \text{(inductive hypothesis)}$$

$$= f(m_j, \pi_A^*(x_j), a_{j+1}) \left[ \alpha_{j+1} \frac{g(\pi_A^*(x_j), a_{j+1})}{f(m_j, \pi_A^*(x_j), a_{j+1}))} m_j + \beta_{j+1} \right] \qquad (Equation\ (13))$$

$$= \alpha_{j+1} g(\pi_A^*(x_j), a_{j+1}) m_j + \beta_{j+1} f(m_j, \pi_A^*(x_j), a_{j+1}))$$

Thus, from the definitions of $f(\cdot), g(\cdot), y(\cdot)$, and $z(\cdot)$,

$$\alpha_j = \begin{cases} \alpha_{j+1} \pi_A^*(x_j) + \beta_{j+1}(2\pi_A^*(x_j) - 1) & a_{j+1} = G \\ \alpha_{j+1}(1 - \pi_A^*(x_j)) + \beta_{j+1}(1 - 2\pi_A^*(x_j)) & a_{j+1} = B \end{cases} \text{ and } \beta_j = \begin{cases} \beta_{j+1}(1 - \pi_A^*(x_j)) & a_{j+1} = G \\ \beta_{j+1}\pi_A^*(x_j) & a_{j+1} = B \end{cases}$$

In both cases, by our inductive hypothesis, $\alpha_{j+1}, \beta_{j+1}$ depend only on $\{\pi_A^*(x_i)\}_{i > j}$ and the actions in this path from $i > j$ to $i = k - 1$. Thus, the claim holds for $\alpha_j, \beta_j$, completing the induction.

Now recall that the denominator of (34) is $\prod_{i \in \mathcal{G}} y(x_i, \pi_A^*(x_i)) \prod_{i \in \mathcal{B} \cup \{k-1\}} z(x_i, \pi_A^*(x_i))$. Note that the policy and sequence of actions is the same as those for the numerator. Thus, because, as shown, the coefficients $\alpha_0, \beta_0$ depend only on $\{\pi_A^*(x_i)\}$ and the path of actions, the same logic can be applied to write the denominator as $\alpha_0 x_0 + \beta_0$ for the same $\alpha_0, \beta_0$. This can be done similarly for the numerator of the second term in (34). Thus, we can reason about the coefficients in (33) as follows:

$$t \left( \prod_{i \in \mathcal{G}} \frac{y(m_i, \pi_A^*(x_i))}{y(x_i, \pi_A^*(x_i))} \right) \left( \prod_{i \in \mathcal{B} \cup \{k-1\}} \frac{z(m_i, \pi_A^*(x_i))}{z(x_i, \pi_A^*(x_i))} \right)$$

$$+ (1 - t) \left( \prod_{i \in \mathcal{G}} \frac{y(n_i, \pi_A^*(x_i))}{y(x_i, \pi_A^*(x_i))} \right) \left( \prod_{i \in \mathcal{B} \cup \{k-1\}} \frac{z(n_i, \pi_A^*(x_i))}{z(x_i, \pi_A^*(x_i))} \right)$$

$$= \frac{t(\alpha_0 m_0 + \beta_0) + (1 - t)(\alpha_0 n_0 + \beta_0)}{\alpha_0 x_0 + \beta_0}$$

$$= \frac{\alpha_0 x_0 + \beta_0}{\alpha_0 x_0 + \beta_0} \qquad \text{(Since } x_0 = t m_0 + (1 - t) n_0)$$

$$= 1$$

Thus, (34) holds, completing the proof. $\qquad \square$

## C.4 ALTRUISTIC MYOPIC POLICY BOUND

**Lemma 9.**

$$\pi_A^0(b) \le \pi_A^*(b) \ \forall b \in [0, 1]$$

Assume by way of contradiction that there exists $b \in [0, 1]$ such that $\pi_A^0(b) > \pi_A^*(b)$. We will show that $V_A^*(b) < V_A^\pi(b)$ for a policy $\pi$ we construct, violating the optimality of $V_A^*(\cdot)$.

Let $\pi$ be such that, when starting at public belief $b$, the policy $\pi$ applies precision $\pi_A^0(b)$ at the current time step and then applies the optimal policy at all future time steps. We now elaborate $V_A^*(b) - V_A^\pi(b)$ by breaking the value function into instantaneous reward $r_A$ and future discounted reward. We will refer to $b^+$ and $b^-$ as the updated beliefs after receiving signals $G$ and $B$, respectively when applying the optimal policy and $b_m^+$ and $b_m^-$ as the same when applying the myopic policy.

$$V_A^*(b) - V_A^\pi(b) = r_A(b, \pi_A^*(b)) - r_A(b, \pi_A^0(b))$$
$$+ \delta y(b, \pi_A^*(b) V_A^*(b^+) + \delta z(b, \pi_A^*(b)) V_A^*(b^-)$$
$$- \delta y(b, \pi_A^0(b)) V_A^*(b_m^+) - \delta z(b, \pi_A^0(b)) V_A^*(b_m^-)$$
$$< \delta y(b, \pi_A^*(b) V_A^*(b^+) + \delta z(b, \pi_A^*(b)) V_A^*(b^-)$$
$$- \delta y(b, \pi_A^0(b)) V_A^*(b_m^+) - \delta z(b, \pi_A^0(b)) V_A^*(b_m^-) \qquad \text{(definition of } \pi_A^0)$$
$$\le 0 \qquad \text{(Theorem 2)}$$

Here, we applied the fact that the instantaneous reward under $\pi_A^0(\cdot)$ must be greater from the definition of the myopic optimal policy.

We then relied on the convexity of $V_A^*(\cdot)$ from Theorem 2 for the future cost. From Equation (13), note that $b_m^- \leq b^- \leq b \leq b^+ \leq b_m^+$. Along with this convexity, this leads to the last step above and completes the proof. ∎

### C.5 PROOF OF THEOREM 3: OPTIMAL ALTRUISTIC POLICY

Consider $V_A^\pi(b)$ with policy $\pi$ applying precision $q$ in the current time step and applying the optimal policy at all future time steps. The $q$ that maximizes $V_A^\pi(b)$ provides the optimum precision starting at public belief $b$. Because the planner never benefits from decreasing precision below the baseline $p$ (see Appendix C.9), we restrict $q$ to $[p, 1]$ without loss of generality.

We will let $b^+$ and $b^-$ denote the positive and negative belief updates possible after receiving a signal with precision $q$. Taking the second derivative of $V_A^\pi(b)$ with respect to $q$ for $q \geq \max(b, 1 - b)$:

$$\frac{\partial^2}{\partial q^2} V_A^\pi(b) = \begin{cases} -\ddot{\beta}(q) + \delta b^2 (1-b)^2 \left[ \frac{\ddot{V}_A^*(b^+)}{y^3(b,q)} + \frac{\ddot{V}_A^*(b^-)}{z^3(b,q)} \right] & q \geq \max(b, 1-b) \\ -\ddot{\beta}(q) & otherwise \end{cases}$$

With the convexity and concavity of $V_A^*(\cdot)$ and $\beta(\cdot)$, respectively, the above shows that $V_A^\pi(b)$ is convex with respect to $q$ on $[\max(b, 1-b), 1]$. Thus, on this interval, it is maximized at one of the two extreme points. For $q \in [0.5, \max(b, 1-b))$, the derivative of $V_A^\pi(b)$ with respect to $q$ is negative; thus, the optimal choice will be the baseline precision $p$ to minimize the cost. Thus, there are three possible optimal precisions: $p$, $\max(b, 1-b)$, and $1$.

Note that the expected value under the latter two candidates is always strictly negative. Under precision $p$, however, the expected value is 0 at $b = 0$ and $b = 1$. This, combined with the non-positivity and symmetry of the value function (Lemma 8) implies the existence of $d_A$ such that $\pi_A^*(b) = p$ for $b \in [0, d_A) \cup (1 - d_A]$. Furthermore, from Theorem 1 and applying Lemma 9, $d_A \in (0, t_M)$ where $t_M$ is defined as in the statement of Theorem 1.

Finally, Theorem 1 and applying Lemma 9, also imply the existence of $t_A \in [d_A, t_M]$ such that $\pi_A^*(b) = 1$ for $b \in (t_A, 1 - t_A)$. This completes the proof. ∎

### C.6 PROOF OF THEOREM 4: MYOPIC BIASED POLICY

**Preliminaries**  Applying the action rule from (2), the instantaneous reward (6) can be written as follows:

$$r_B(b, q) = \begin{cases} -\beta(|q - p|) - Cz(b, q) & q \geq \max(b, 1-b) \\ -\beta(|q - p|) - C & b < 1 - q \\ -\beta(|q - p|) & b > q \end{cases} \tag{44}$$

Note that $z(b, q) = b + q - 2bq$ increases with respect to $q$ when $b < 0.5$ and decreases with respect to $q$ when $b > 0.5$.

At $b = 0.5$, $z(b, q) = z(0.5, q) = 0.5$ regardless of the precision chosen. Furthermore, at this public belief, $q \geq \max(b, 1 - b)$ because $q \geq 0.5$ for any $q$, falling in the first case of (44). In this case, the second term of $r_B(b, q)$ is unchanging with respect to $q$, so the planner seeks to maximize only the first term, which is accomplished at $q = p$.

We now prove a series of lemmas (10-14) which will assist us in our proof of Theorem 4.

**Lemma 10.**
*If $\pi_B^0(b) \geq \max(b, 1 - b)$, then $\pi_B^0(b) \in \{\max(b, 1 - b), p, 1\}$.*

*Proof.*  The reward is differentiable everywhere except $q = p$. Thus, we can write its derivative with respect to the chosen precision as follows (except at $q = p$):

$$\frac{\partial r_B(b, q)}{\partial q} = \begin{cases} -\operatorname{sign}(q - p)\dot{\beta}(|q - p|) \\ -C(1 - 2b) & q \geq \max(b, 1-b) \\ -\operatorname{sign}(q - p)\dot{\beta}(|q - p|) & otherwise \end{cases} \tag{45}$$

In the former case, we can write the second derivative of the reward with respect to $q$ as follows, for $q \neq p$:

$$\frac{\partial^2}{\partial q^2} r_B(b, q) = -\ddot{\beta}(|q - p|)$$

Because $\beta(\cdot)$ is concave, this expression is non-negative, and $r_B(b, q)$ is convex with respect to $q$. Thus, the reward is maximized at one of the extreme points $\{\max(b, 1 - b), 1\}$ or $p$ because $r_B(b, q)$ is not differentiable at $q = p$. □

**Lemma 11.**
For $b \in [0, 1 - p)$, $\pi_B^0(b) \in \{p, 1 - b\}$.

*Proof.* Because $p > 0.5$, for $b \in [0, 1 - p)$, $\max(b, 1 - b) = 1 - b > p$.

First, consider the case when $\pi_B^0(b) < \max(b, 1 - b) = 1 - b$. From (44), in such instances, the reward is maximized by minimizing $|q - p|$. Thus, the optimal precision lower than $\max(b, 1 - b)$ is $p$.

Now consider $\pi_B^0(b) \geq \max(b, 1 - b) = 1 - b$. By Lemma 10, $\pi_B^0(b) \in \{1 - b, p, 1\}$. What remains is to rule out $\pi_B^0(b) = 1$. To do so, we compare $r_B(b, 1)$ and $r_B(b, 1 - b)$. Both fall under the first case of (44). Because $\beta(\cdot)$ is increasing in its argument and $p < 1 - b \leq 1$, the first term of the reward from (44) is greater for precision $1 - b$ than for precision 1. Since $z(b, q)$ is increasing with respect to $q$ for $b < 0.5$, $r_B(b, 1) \leq r_B(b, \max(b, 1 - b))$ so we can rule out precision 1. This completes the proof. □

**Lemma 12.**
For $b \in (p, 1]$, $\pi_B^0(b) = p$.

*Proof.* The reward is strictly negative for any $q \neq p$ because of the term $-\beta(|q - p|)$. For $b \in (p, 1]$ and $q = p$, $p < \max(b, 1 - b) = b$, leading to the third case of (44) and yielding reward 0. Thus, the optimal precision is $p$. □

**Lemma 13.**
For $b \in [1 - p, 0.5]$, $\pi_B^0(b) \in \{p, 1 - b\}$.

*Proof.* When $p = 0.5$, the interval $[1 - p, 0.5]$ contains only $\{0.5\}$. At $b = 0.5$ $p = 1 - b$, thus $\pi_B^0(b) = p = 1 - b = 0.5$. We deal with $p > 0.5 \geq b$ for the remainder of the proof.

For $b \in [1 - p, 0.5]$, $\max(b, 1 - b) = 1 - b \leq p$. Thus, from the first case of (44), precision $p$ yields reward $r_B(b, p) = -Cz(b, p)$. We now consider two cases: $q < \max(b, 1 - b)$ and $q \geq \max(b, 1 - b)$.

Suppose $q < \max(b, 1 - b) = 1 - b$. This leads to the second case of (44) and yields reward $-\beta(|q - p|) - C$. Furthermore, $z(b, q)$ is increasing with respect to $q$ for $b < 0.5$ and takes values between $z(b, 0.5) = 0.5$ and $z(b, 1) = 1 - b$. Thus, $z(b, q) < 1$ for all $q$. Therefore, the reward from policy $q < 1 - b$ is strictly lower than $r_B(b, p)$, and we can rule out this case.

When $q \geq \max(b, 1 - b)$, applying Lemma 10 yields that the remaining candidate precisions are $\{1 - b, p, 1\}$. We rule out precision 1 by comparing it to precision $p$. The first term of the reward from (44) is greater for precision $p$ than for precision 1. Since $z(b, q)$ is increasing with respect to $q$ for $b < 0.5$, the second term is also greater for precision $p$, and we can rule out precision 1. This completes the proof. □

We now define the notion of $\epsilon$-optimal policies.

**Definition 1.** *$\epsilon$-optimal policies*

*An $\epsilon$-optimal policy $\pi^\epsilon(b)$ is any policy such that the following holds*

$$V^*(b) \leq V^{\pi^\epsilon}(b) + \epsilon$$

*for $\epsilon > 0$.*

Note that from the definition, any optimal policy is also $\epsilon$-optimal for any $\epsilon > 0$. Thus, $\pi_B^0(b)$ is also $\epsilon$-optimal

Following this definition, let $\pi_B^{\epsilon,0}(b)$ denote an $\epsilon$-optimal myopic policy for the biased planner.

**Lemma 14.**
For $b \in (0.5, p]$ and any $\epsilon > 0$, there are two possibilities:

1. $\pi_B^0(b) \in \{p, 1\}$, or

2. $\pi_B^{\epsilon,0}(b) = b - \epsilon$

*Proof.* For $b \in (0.5, p]$, $\max(b, 1 - b) = b \leq p$. We consider two cases: $q \geq \max(b, 1 - b)$ or $q < \max(b, 1 - b)$.

In the first case, Lemma 10 yields that $\pi_B^0(b) \in \{b, p, 1\}$. We rule out precision $b$ by comparing it to precision $p$. The first terms of the reward from (44) is greater for precision $p$. Furthermore, since $z(b, q)$ is decreasing with respect to $q$ for $b > 0.5$, the second term of the reward is also greater for precision $p$. Thus, we can rule out precision $b$. This yields the first claim.

In the latter case, $q < \max(b, 1 - b) = b$. Therefore, (1) since $q \geq 0.5$, $q \in [0.5, b)$, and (2) the third case of (44) applies leading to reward $-\beta(|q - p|)$. Thus, the reward $r_B(b, q)$ is an increasing function of $q \in [0.5, b)$ because $0.5 < b \leq p$. This leads directly to the second claim. $\qquad \square$

**Remark 3.** *As noted in Puterman (1990) Section 2.3.1, an optimal policy may not exist for infinite action spaces because the supremum may not be attained. When an optimal policy does not exist, we instead seek $\epsilon$-optimal policies, i.e., policies that yield reward within $\epsilon > 0$ of the supremum. In particular, this is true for case 2 of Lemma 14.*

**Proof of Theorem 4**  We consider the myopic biased policy in 4 cases: $b \in [0, 1 - p)$, $b \in [1 - p, 0.5)$, $b \in (0.5, p]$, and $b \in (p, 1]$, making use of the fact that $p \geq 0.5$. Based on Lemmas 11-14, these cases have the following candidate policies: $\{p, 1 - b\}$, $\{1 - b, p\}$, $\{1, p, b - \epsilon\}$, and $\{p\}$, respectively.

*Case 1:* $b \in [0, 1 - p)$, $\pi_B^0(b) \in \{p, 1 - b\}$ (Lemma 11)

We will show the existence of a threshold $t_1$ such that $\pi_B^0(b) = p$ for $b \in [0, t_1]$ and $\pi_B^0(b) = 1 - b$ for $b \in (t_1, 1 - p)$.

We can express the reward under the two candidate policies as follows by applying (44):

$$r_B(b, p) = -C$$
$$r_B(b, 1 - b) = -\beta(1 - b - p) - C(1 - 2b(1 - b))$$

The derivative of $r_B(b, 1 - b)$ with respect to $b$ is

$$\frac{\partial}{\partial b} r_B(b, 1 - b) = \dot{\beta}(1 - p - b) + 2C(1 - 2b)$$

which is positive as $\beta(\cdot)$ is increasing and $b < 1 - p \leq 0.5$. Furthermore, when evaluated at $b = 0$ and $b = 1 - p$ we obtain

$$r_B(0, 1) = -\beta(1 - p) - C < r_B(b, p)$$
$$r_B(1 - p, p) = -C(1 - 2p(1 - p)) > r_B(b, p)$$

Thus, $r_B(b, 1 - b)$ is increasing on $[0, 1 - p)$ with $r_B(b, 1 - b) < r_B(b, p)$ for $b = 0$ and vice versa for $b = 1 - p$. Because $\beta(\cdot)$ is continuous, so are both reward functions. Thus, there exists $t_1 \in (0, 1 - p)$ such that $\pi_B^0(b) = p$ for $b \in [0, t_1]$ and $\pi_B^0(b) = 1 - b$ for $b \in (t_1, 1 - p)$.

*Case 2:* $b \in [1 - p, 0.5]$, $\pi_B^0(b) \in \{1 - b, p\}$ (Lemma 13)

If $p = 0.5$, this interval contains only $\{0.5\}$, and, for $b = 0.5$, $1 - b = p$. Thus, the two candidate policies are equivalent, and the optimal policy is $\pi_B^0(b) = p$. For the remainder of this case, we consider $p > 0.5$.

We now prove the existence of $t_2, t_3$ such that $\pi_B^0(b) = 1 - b$ for $b \in [t_2, t_3)$, and $\pi_B^0(b) = p$ for $b \in [1 - p, t_2) \cup [t_3, 0.5]$.

We can express the reward under the candidate policies as follows by applying (44):

$$r_B(b, 1 - b) = -\beta(p - 1 + b) - Cz(b, 1 - b)$$
$$r_B(b, p) = -Cz(b, p)$$

This leads to the following condition for the optimal policy to be $1 - b$:

$$r_B(b, 1 - b) \geq r_B(b, p)$$
$$\beta(p - 1 + b) \leq C[z(b, p) - z(b, 1 - b)]$$
$$\beta(p - 1 + b) \leq C(p - 1 + b)(1 - 2b) \qquad (46)$$

The LHS of (46) is a concave, increasing function of $b$ on [1-p,0.5]. The RHS of (46) is a concave parabola that increases in the first half of $[1 - p, 0.5)$ and decreases in the second half.

Now consider the two halves of the interval $[1 - p, 0.75 - 0.5p]$ and $[0.75 - 0.5p, 0.5)$. In the first half, the LHS and RHS of (46) are increasing and concave, with the RHS also being a parabola. Thus, in the first half,

the two intersect at most two points (one of which is at $b = 1 - p$). In the second half, the LHS of (46) is increasing while the LHS is decreasing. Thus, they intersect at most once. Let the two possible intersections for $b > 1 - p$ be $t_2$ and $t_3$ with $t_2 \leq t_3$. Thus, what remains is to determine the optimal on each of $(1 - p, t_2)$, $(t_2, t_3)$, and $(t_3, 0.5)$. At $b = 0.5$, $r_B(b, p) > r_B(b, 1 - b)$ because $p > 0.5$, thus $\pi_B^0(b) = p$ for $b \in (t_3, 0.5)$. Subsequently, $\pi_B^0(b) = 1 - b$ for $(t_2, t_3)$, and $\pi_B^0(b) = p$ for $(1 - p, t_2)$.

_Case 3:_ $b \in (0.5, p], \pi_B^{\epsilon;0}(b) \in \{1, p, b - \epsilon\}$ (Lemma 14)

In this case, we show the existence of $0.5 \leq t_4 \leq t_5 \leq p$ such that the optimal precisions on intervals $(0.5, t_4]$, $(t_4, t_5)$, and $[t_5, p)$ are $p$, $1$, and $b - \epsilon$, respectively.

We can express the reward under the candidate policies as follows by applying (44):

$$r_B(b, 1) = -\beta(1 - p) - C(1 - b)$$
$$r_B(b, p) = -Cz(b, p)$$
$$r_B(b, b - \epsilon) = -\beta(p - b + \epsilon)$$

When evaluated at $b = p$, $r_B(b, b - \epsilon)$ is arbitrarily close to 0, and, therefore, $b - \epsilon$ is the optimal precision on some interval $(t_5, p]$ for $0.5 \leq t_5 < p$.

Now, note that $r_B(b, 1)$ and $r_B(b, p)$ are linear functions in $b$ while $r_B(b, b - \epsilon)$ is convex in $b$ (from the concavity of $\beta$). Thus, $r_B(b, b - \epsilon) - r_B(b, p)$ and $r_B(b, b - \epsilon) - r_B(b, 1)$ are convex and, as a result, have convex level sets. This implies that the set on which both of these expressions are positive, i.e., the set for which $b - \epsilon$ is the optimal policy, must also be convex. Therefore, $b - \epsilon$ cannot be optimal on any subset of $(0.5, t_5)$.

What remains is to compare precisions 1 and $p$ on the interval $(0.5, t_5]$. To do so, we inspect the values of $r_B(b, 1)$ and $r_B(b, p)$ at 0.5 and their derivatives.

$$r_B(0.5, 1) = -\beta(1 - p) - 0.5C$$
$$r_B(0.5, p) = -0.5C$$

Because $r_B(0.5, 1) < r_B(0.5, p)$ there exists interval $(0.5, t_4)$ with $t_4 \leq t_5$ such that $\pi_B^0(b) = p$. Furthermore, because both are linear in $b$, there is at most one crossover point where their ordering changes. Thus, if $t_4 < t_5$, then $pi_B^0(b) = 1$ on $(t_4, t_5)$.

_Case 4:_ $b \in (p, 1], \pi_B^0(b) = p$

This follows directly from Lemma 12 and completes the proof of Theorem 4. ∎

## C.7   BIASED VALUE FUNCTION MONOTONICITY

**Lemma 15.** _For any $b_1, b_2 \in [0, 1]$ such that $b_1 \leq b_2$,_

$$V_B^*(b_1) \leq V_B^*(b_2)$$

_Proof._ Applying (44), the derivative of the biased reward with respect to $b$ can be written as follows:

$$\frac{\partial}{\partial b} r_B(b, q) = \begin{cases} -C(1 - 2q) & q \geq \max(b, 1 - b) \\ 0 & \text{otherwise} \end{cases} \tag{47}$$

Because $q \geq 0.5$ and $z(b, q) \in [0, 1]$, it follows that $r_B(b, q)$ is increasing with respect to $b$.

Next, note that the Bayesian nature of belief updates implies that the belief process is a martingale. That is, for any fixed $q_i$, $\mathbb{E}[b_{i+1}] = b_i$. To demonstrate this consider two cases: (1) $q_i < \max(b_i, 1 - b_i)$, and (2) $q_i \geq \max(b_i, 1 - b_i)$.

In the former case, it follows from (3) that $b_{i+1} = b_i$, satisfying the claim.

In the latter case, (13) gives the distribution of $b_{i+1}$ which yields the same.

The monotonicity of $V_B^*(\cdot)$ then follows from Proposition 5 of Smith & McCardle (2002). □

## C.8   PROOF OF THEOREM 5: OPTIMAL BIASED POLICY

We first prove that the optimal value function is continuous in the following lemma:

**Lemma 16.** _Biased value function continuity_

$$V_B^*(\cdot) \text{ is continuous w.r.t. } b.$$

*Proof.* First consider the one-stage value function defined as $V_B^0(b) = \sup_{q \in [0.5,1]} r_B(b,q)$. From Theorem 4, we know that $\pi_B^{\epsilon,0}(b) \in \{1-b, 1, b-\epsilon, p\}$. Thus, we can equivalently write the one-stage value function as follows:

$$V_B^0(b) = \sup_{\epsilon > 0} r_B \left( b, \pi_B^{\epsilon,0}(b) \right)$$
$$= \sup_{\epsilon > 0} \max_{q \in \{p, 1-b, 1, b-\epsilon, p\}} r_B(b,q) \tag{48}$$

Using the above expression, we now argue that $V_B^0(\cdot)$ is continuous on $b \in [0,1]$. First, consider $b \leq 0.5$. From Theorem 4, an $\epsilon$-optimal policy applies precision $1-b$ or $p$ for $b \leq 0.5$. In either case, the instantaneous reward will not depend upon $\epsilon$, and the supremum in (48) can be omitted, leaving $V_B^0(b) = \max_{q \in \{1-b, p\}} r_B(b,q)$. For $b \leq 0.5$, $r_B(b, 1-b)$ falls in case one of (44) and is thus continuous with respect to $b$. On the same interval, $r_B(b,p)$ has one discontinuity at $b = 1-p$ where it switches from case two to case one of (44). At $b = 1-p$, Theorem 4 shows that the optimal myopic policy switches from $1-b$ to $p$. Both policies fall in the first case of (44), and, at $b = 1-p$, $p = 1-b$, so the controls are equal. Thus, the left and right limits coincide with the optimal value, and $V_B^0(\cdot)$ is continuous.

For $b \geq 0.5$, Theorem 4 gives the candidate $\epsilon$-optimal policies $q \in \{1, b-\epsilon, p\}$. When the inner maximum in (48) is attained for $q = b-\epsilon$, the resulting value is $V_B^0(b) = \sup_{\epsilon > 0} r_B(b, b-\epsilon)$. From the third case of (44), we can rewrite this as $V_B^0(b) = \sup_{\epsilon > 0} -\beta(|b - \epsilon - p|)$. Since $b - \epsilon$ is only selected for $b \leq p$ (Theorem 4), this reduces to $V_B^0(b) = -\beta(|p - b|)$. Thus, exchanging the order of the maximum and supremum in (48), we obtain $V_B^0(b) = \max\{r_B(b,1), -\beta(|p-b|), r_B(b,p)\}$ for $b \geq 0.5$. For the first expression in this maximum, $r_B(b,1)$ falls into case one of (44) and is thus continuous. The continuity of $-\beta(|p-b|)$ follows from the continuity of $\beta(\cdot)$. For $r_B(b,p)$, since $b \geq 0.5$, the first case of (44) applies for $b \leq p$ and the third case applies for $b > p$. Thus, $r_B(b,p)$ is potentially discontinuous at $b = p$. Therefore, what remains is to show the continuity of $V_B^0(\cdot)$ at $b = p$.

At $b = p$, the myopic $\epsilon$-optimal policy switches from $q = b-\epsilon$ to $q = p$ by Theorem 4 with $\pi_B^{\epsilon,0}(p) = b - \epsilon$. Thus, for $b = p$, the inner maximum of (48) is attained at $q = b - \epsilon$ and, from the third case of (44), $V_B^0(p) = \sup_{\epsilon > 0} -\beta(\epsilon) = 0$. Similarly, from Theorem 4, for each $\epsilon > 0$, there exists $t_5 < p$ such that on $[t_5, p]$ the inner maximum of (48) is attained by $q = b - \epsilon$. On this interval, $V_B^0(b) = \sup_{\epsilon > 0} -\beta(|p - b + \epsilon|) = -\beta(|p - b|)$. Therefore, $\lim_{b \uparrow p} V_B^0(b) = 0$. For $b > p$, the myopic optimal policy is $q = p$ from Theorem 4, and this yields reward $V_B^0(b) = 0$ for all $b > p$ from the third case of (44). This proves the continuity of $V_B^0(\cdot)$ at $b = p$ and completes the proof that $V_B^0(\cdot)$ is continuous on $[0,1]$.

Applying Theorem 4.2 of Hinderer (2005), then yields the continuity of $V_B^*(\cdot)$. □

**Remark 4.** *Given the continuity of $V_B^*(\cdot)$ with respect to public belief and the fact that the control variable lies in the compact set $[0.5, 1]$, one might conclude that the optimal is guaranteed to exist at all public beliefs. However, there is an important subtlety which prevents such a conclusion: while $V_B^*(\cdot)$ is continuous with respect to public belief, it need not be continuous with respect to precision. Specifically, $b_{i+1}$ is a function of both $b_i$ and $q_i$ (see Equation (3)); thus, $\mathbb{E}[V_B^*(b_{i+1})]$ may be denoted as $\phi(b_i, q_i)$. Now, $b_{i+1}$ need not be continuous with respect to $q_i$, thus $\phi(\cdot)$ may not be continuous with respect to $q_i$. Thus, there may be regimes in which the optimal value is not attained by any policy. Indeed, we will show that such cases exist.*

**Proof of Theorem 5** We now leverage Lemma 16 to prove Theorem 5. We again let $z(b,q) = b + q - 2bq$ be the probability of an agent receiving signal $B$ with precision $q$ and prior public belief $b$ and $y(b,q) = 1 - z(b,q)$ be the corresponding probability of the agent receiving signal $G$. We abuse notation to allow $V_B(b,q)$ to be the expected utility when beginning at public belief $b$, applying precision $q$ in the current time step, and then following the optimal policy $\pi_B^*(\cdot)$ in all future steps. We can express $V_B(b,q)$ as follows by applying (13), (3), and (44):

$$V_B(b,q) = \begin{cases} -\beta(|q-p|) - Cz(b,q) \\ +\delta y(b,q) V_B^* (f(b,q,G)) \\ +\delta z(b,q) V_B^* (f(b,q,B)) & q \geq \max(b, 1-b) \\ -\beta(|q-p|) \\ -C\mathbf{1}(b < 0.5) \\ +\delta V_B^*(b) & \text{otherwise} \end{cases} \tag{49}$$

Consider the latter case of (49) when $q < \max(b, 1-b)$. Here, the only term affected by the precision policy is $-\beta(|q-p|)$, which is maximized by choosing $q$ as close to $p$ as possible. Thus, if the optimal precision is less than $\max(b, 1-b)$, then it will be as close to $p$ as possible.

We now consider four cases: (1) $b \in [0, 1-p)$, (2) $b \in [1-p, 0.5)$, (3) $b \in [0.5, p]$, and (4) $b \in (p, 1]$.

_Case 1_: $b \in [0, 1 - p)$, claims (A) and (B)

In this case, we will prove the existence of $t_1$ such that $\pi_B^*(b) = p$ for $b \leq t_1$. First note that if $\pi^*(b) < \max(b, 1 - b)$, we fall into the second case of Equation (49). Here, the only term affected by the precision policy is $-\beta(|q - p|)$, which is maximized by minimizing $|q - p|$. Because $p \in [0.5, 1]$, when $b \in [0, 1 - p)$, $p < \max(b, 1 - b)$. Thus, for $b \in [0, 1 - p)$, if $\pi_B^*(b) < \max(b, 1 - b)$, then $\pi_B^*(b) = p$.

Now, we evaluate the value function at belief $b = 0$. At $b = 0$, the state of the world is known to be $B$ and, thus, no signal, regardless of its precision, will move the public belief. This can be seen mathematically by substituting $b = 0$ into the updated belief expressions in (13). Furthermore, at $b = 0$, if $q \geq \max(b, 1 - b)$, then $q = 1$. As argued above, if the precision is less than $\max(b, 1 - b)$, the optimal precision is $q = p$. We can compare the resultant value under these two candidate policies as follows from Equation (49):

$$V_B(0, 1) = -\beta(1 - p) - C + \delta V_B^*(0) \tag{50}$$

$$V_B(0, p) = -C + \delta V_B^*(0) \tag{51}$$

Because $V_B(0, 1) < V_B(0, p)$, $\pi_B^*(0) = p$ and $V_B^*(0) = V_B(0, p)$. Thus, from Equation (51), $V_B^*(0) = \frac{-C}{1-\delta}$. By Lemma 15, this is a lower bound for the value function at any public belief. Therefore, and since $V_B^*(b) \leq 0$, $|V_B^*(b)| \leq \frac{C}{1-\delta}$ for all $b \in [0, 1]$.

We now show that there exists an interval $[0, \epsilon)$ with $\epsilon > 0$ on which the optimum of the first case of Equation (49) is less than the optimum of the second case of Equation (49). Thus, in this interval the optimum control lies in the second case of Equation (49) so $\pi_B^*(b) < \max(b, 1 - b)$. It follows from our initial argument earlier in this case that $\pi_B^*(b) = p$ in this interval, which would complete the proof.

We define function $g(\cdot)$ to be the maximum of the first case of the RHS of Equation (49):

$$\begin{aligned} g(b) = \max_{q \geq \max(b, 1-b)} \big[ &- \beta(|q - p|) - Cz(b, q) \\ &+ \delta y(b, q) V_B^* (f(b, q, G)) \\ &+ \delta z(b, q) V_B^* (f(b, q, B)) \big] \end{aligned} \tag{52}$$

For $b \in (0, 1)$, $f(b, q, G)$ and $f(b, q, B)$ are both continuous with respect to $q$ from Equation (13). Thus, the maximum in Equation (52) must exist because it is of a function which is continuous with respect to $q$ over a compact set $q \in [\max(b, 1 - b), 1]$.

We now take the limit of $g(\epsilon)$ as $\epsilon$ tends to 0. Because $q \in [\max(\epsilon, 1 - \epsilon), 1]$, when $\epsilon \to 0$, $q \to 1$ (inside the maximum in Equation (52)). We now reason about each of the four terms in Equation (52). As $\epsilon \to 0$, $q \to 1$ so that 1) from the continuity of $\beta(\cdot)$, $\beta(|q - p|) \to \beta(1 - p)$, 2) from the continuity of $z(\cdot)$, $z(\epsilon, q) \to 1$, $y(\epsilon, q) \to 0$. Thus, the second term goes to $C$. Because $V_B^*(\cdot)$ is bounded and $y(\epsilon, q) \to 0$, the third term goes to 0. Finally, because $z(\epsilon, q) \to 1$ and $f(\epsilon, q, B) \to 0$ and since $V_B^*(\cdot)$ is continuous from Lemma 16, the last term approaches $\delta V_B^*(0)$. Thus,

$$\lim_{\epsilon \to 0} g(\epsilon) = -\beta(1 - p) - C + \delta V_B^*(0) \tag{53}$$

Applying precision $p$, however, yields the following from the second case of Equation (49):

$$\lim_{\epsilon \to 0} V_B(\epsilon, p) = \lim_{\epsilon \to 0} [-\beta(p - p) - C + \delta V_B^*(\epsilon)] \tag{54}$$

$$= -C + \delta V_B^*(0) \tag{55}$$

The last equality follows from the continuity of $V_B^*(\cdot)$ from Lemma 16.

Note that the expression in Equation (53) is strictly less than that in Equation (55). Thus, there exists an interval $[0, \epsilon)$ with $\epsilon > 0$ on which the optimum of the first case of Equation (49) is less than the optimum of the second case of Equation (49). The result follows.

_Case 2_: $b \in [1 - p, 0.5)$, Claim (C)

We now argue that for $b_i \in [1 - p, 0.5)$, $\pi^*(b_i) \geq 1 - b_i$, yielding claim (C). Consider a stationary policy $\pi(\cdot)$ such that $\pi(b_i) < 1 - b_i$. Note that, from Equation (3), the public belief will remain unchanged under such a policy. Thus, the same control will be applied at every future time step, and, from Equation (2), every agent will select action $B$. Using the facts that $1 - b_i \leq p$ in this belief range, $\mathbb{P}(a_i = B | b_i, \pi(b_i)) = 1$, and $b_{i+1} = b_i$, we can apply Equation (6) to write the expected total discounted utility under such a policy as follows:

$$V_B^\pi(b_i) = -\beta(p - \pi(b_i)) - C + \delta V_B^\pi(b_i)$$

Thus, if this policy is applied, it will result in expected value $V_B^\pi(b_i) = \frac{-\beta(p - \pi(b_i)) - C}{1 - \delta}$. That is, the cost of every agent choosing action $B$ and continuing to apply precision $\pi(b_i)$. This is strictly lower than the value

function lower bound of $\frac{-C}{1-\delta}$ established using Lemma 15 in Case 1 because $\pi(b_i) < 1 - b_i \leq p$. Therefore, if an optimal policy exists at this point, it must be the case that $\pi_B^*(b_i) \geq 1 - b_i$.

The remaining candidate policies are $q_i \in [1 - b_i, 1]$. Applying Bellman's principle we can write the following:

$$V_B^*(b_i) = \sup_{q_i \in [1-b_i, 1]} r_B(b_i, q_i) + \delta \mathbb{E}[V_B^*(b_{i+1})] \tag{56}$$

Because $q_i \geq \max(b_i, 1 - b_i)$ and $b \in [1 - p, 0.5) \subset (0, 1)$, Equation (13) and Equation (3) imply that $b_{i+1} = \frac{b_i q_i}{y(b_i, q_i)}$ or $b_{i+1} = \frac{b_i(1-q_i)}{z(b_i, q_i)}$. Both of these expression are continuous with respect to $q_i$ and $V_B^*(\cdot)$ is continuous by Lemma 16, thus the RHS of Equation (56) inside the supremum is continuous with respect to $q_i$. Therefore, in Equation (56), the supremum over a compact set $q_i \in [1 - b, 1]$ of a function which is continuous with respect to $q_i$ must be attained, implying the existence of an optimal policy. Thus, $\pi_B^*(b_i) \geq 1 - b_i$.

_Case 3:_ $b \in [0.5, p]$ Claims (D) and (E)

For $b = 0.5$, $q \geq b$ as we assumed $q \in [0.5, 1]$. This places us in the first case of Equation (49). Furthermore, $\beta(\cdot), y(b, q), z(b, q), f(b, q, G)$, and $f(b, q, B)$ are all continuous with respect to $q$ for $b = 0.5$. Thus, the supremum over $q \in [0.5, 1]$ of $V(b, q)$ must be attained as it is of a continuous function of $q$ over a compact set of $q$. Therefore, an optimal policy must exist at $b = 0.5$.

Now consider $b \in (0.5, p]$ and a stationary policy $\pi'(\cdot)$ such that $\pi'(b) < b$. From Equation (3) and Equation (2), applying such a policy will result in public belief remaining unchanged, every agent selecting action $G$, and the same control being applied at every future time step. Using the facts that $b_i \leq p$, $\mathbb{P}(a_i = G | b_i, \pi'(b_i)) = 1$, and $b_{i+1} = b_i$, we can apply Equation (6) to write the total expected discounted utility under such a policy as follows:

$$V_B^{\pi'}(b_i) = -\beta(p - \pi'(b_i)) + \delta V_B^{\pi'}(b_i)$$

Thus, the policy results in expected value $V_B^{\pi'}(b_i) = \frac{-\beta(p-\pi'(b_i))}{1-\delta}$. Note that, in this case, $V_B^{\pi'}(b_i)$ is an increasing function of $\pi'(b_i)$. Thus, the supremum of $V^*(\pi')_B(b_i)$ over $\pi'(b_i) \in (0.5, b)$ is not attained.

We now consider two cases: (1) $\max_{q_i \in [b_i, 1]} V(b_i, q_i) \geq \sup_{\pi'(b_i) \in (0.5, b_i)} \frac{-\beta(p-\pi'(b_i))}{1-\delta}$, and (2) the complement. In the former case, we can argue as we did for $b = 0.5$ that the supremum must be attained and an optimal policy exists such that $\pi_B^*(b_i) \geq b_i$ (claim (D)). In the latter case, as argued above, no optimal policy exists. Instead, for sufficiently small $\epsilon > 0$, there exists an $\epsilon$-optimal policy such that $\pi_B^\epsilon(b_i) = b_i - \epsilon$.

What remains is to show that there exists $t_2 \in (0.5, p)$ such that for $b_i \in (t_2, p]$ we fall into the latter case, yielding claim (E). We define functions $m(\cdot)$ and $n(\cdot)$ as follows corresponding to the resultant value from choosing precisions less than $b$ or greater than or equal to $b$, respectively:

$$m(b) = \sup_{\pi'(b) \in [0.5, b)} \frac{-\beta(p - \pi'(b))}{1 - \delta}$$
$$n(b) = \max_{q \geq b} V(b, q)$$

Taking the limit of $m(\cdot)$ as $b \to p$ yields:

$$\lim_{b \to p} m(b) = 0 \tag{57}$$

When $q \geq b$ we fall into the first case of Equation (49). Thus we can expand $n(b)$ as follows:

$$n(b) = \max_{q \geq b} \big[ -\beta(|q - p|) - Cz(b, q)$$
$$+ \delta y(b, q) V_B^*(b, q)$$
$$+ \delta z(b, q) V_B^*(b, q) \big] \tag{58}$$

We will argue that, for $b \in (0.5, p]$ and $q \geq b$, $n(b) \leq -C(1 - p)$. Since $p$ is a fixed parameter strictly less than 1, this means from Equation (57) that $n(b) \leq -C(1 - p) < \lim_{b \to p} m(b)$, which implies the existence of an open interval $(t_2, p)$ where an optimal policy does not exist, yielding claim (F). We now argue that, for $b \in (0.5, p]$ and $q \geq b$, $n(b) \leq -C(1 - p)$. Note that every term of Equation (58) is bounded above by 0, so it suffices to show that any one term, say the second term $-Cz(b, q)$, is upper bounded by $-C(1 - p)$. First note that $z(b, q) = q(1 - 2b) + b$ so $z(b, q)$ is decreasing in $q$ since $b > 0.5$. Thus, $z(b, q) \geq z(b, 1) = 1 - b \geq 1 - p$. The last inequality follows because $b \in (0.5, p]$. The claim follows since $p < 1$.

Along with Equation (57), this implies the existence of an open interval $(t_2, p)$ where an optimal policy does not exist, yielding claim (F).

_Case 4:_ $b \in (p, 1]$ Claim (A)

For $b \in (p, 1]$, consider applying stationary policy $\pi(\cdot)$ such that $\pi(b) = p$. Because $b > p$, Equation (3) and Equation (2) imply that such a policy results in public belief remaining unchanged, every agent selecting action $G$, and the same control being applied at every future time step. We can apply Equation (6) to write

$$V_B^\pi(b) = -\beta(p - p) + \delta V_B^\pi(b) = \delta V_B^\pi(b)$$

Thus, $V_B^\pi(b) = 0$. Since this is also an upper bound of the optimal value function for any $b$, $\pi_B^*(b) = p$. ∎

## C.9 SOCIAL WELFARE MONOTONICITY

**Lemma 17.** *Precision Monotonicity of Social Welfare*

*If $\pi_1(b) \leq \pi_2(b) \ \forall b \in [0, 1]$, for policies $\pi_1(\cdot), \pi_2(\cdot)$, then*

$$W^{\pi_1}(b) \leq W^{\pi_2}(b) \ \forall b \in [0, 1]$$

*Proof.* It is clear that the one-step welfare $-C \min(b_i, 1 - b_i, 1 - \pi(b_i))$ is increasing in $\pi(b_i)$.

Furthermore, note that the convexity of social welfare with respect to public belief follows from the same argument as in the proof of Theorem 2. This is because social welfare is equivalent to the altruistic value function in the special case where $\beta(\cdot) = 0$.

We can then follow the same argument as in the proof of Lemma 9 (Appendix C.4) to show that increasing precision increases welfare. □

# D EXTENSION TO HETEROGENEOUS AGENT PREFERENCES

Here we argue that, for the altruistic planner, the problem we consider is equivalent to one that involves two types of individuals with differing preferences.

Assume that there are two types of agents $G$ and $B$ with the type of agent $i$ denoted $t_i$. The first type receives unit reward when $a_i = \omega$ and no reward otherwise. The second receives unit reward when $a_i \neq \omega$ and no reward otherwise. Further, assume that all agents know their own type and the types of all other agents. All other aspects of the model remain the same.

The private belief update of agent $i$ can be expressed as below. This remains unchanged because it only depends on the signal structure, i.e., $\mathbb{P}(s_i = \omega) = q_i$.

$$\tilde{b}_i = \mathbb{P}(\omega = G | \mathcal{H}_i, q_i, s_i) = \begin{cases} \frac{q}{1 + 2b_i q_i - b_i - q_i} b_i & s_i = G \\ \frac{1-q}{b_i + q_i - 2b_i q_i} b_i & s_i = B \end{cases} \tag{59}$$

Using Equation (59), agent $i$'s action can be written as follows with superscript indicating the type of agent $i$ and ! indicating negation:

$$a_i^G = \begin{cases} s_i & 1 - q_i \leq b_i \leq q_i \\ G & q_i < b_i \\ B & b_i < 1 - q_i \end{cases}$$

$$a_i^B = \begin{cases} !s_i & 1 - q_i \leq b_i \leq q_i \\ B & q_i < b_i \\ G & b_i < 1 - q_i \end{cases}$$

Because we assume the type of each agent is known to all, when $q_i \geq \max(b_i, 1 - b_i)$, all agents can perfectly infer the private signal of agent $i$. If $q_i < \max(b, 1 - b_i)$, the action of agent $i$ is uninformative.

Thus, after observing agent $i$'s type $t_i$, precision $q_i$, and action $a_i$, the public belief is updated as follows:

$$b_{i+1} = f(b_i, q_i) = \begin{cases} \tilde{b}_i & 1 - q_i \leq b_i \leq q_i \\ b_i & \text{o.w.} \end{cases} \tag{60}$$

In this setting, an altruistic planner seeks to induce type-$G$ agents to take action $\omega$ and type-$B$ agents to take action $!\omega$ as these choices align with the personal utility functions of the agents.

Thus, we can express the planner's instantaneous reward as follows with superscript denoting the type of agent $i$:

$$r^G(b_i, q_i) = -\beta(|q_i - p|) - C\,\mathbb{P}(a_i \neq \omega)$$

$$= \begin{cases} -\beta(|q_i - p|) - C(1 - q_i) & q_i \geq \max(b_i, 1 - b_i) \\ -\beta(|q_i - p|) - C\min(b_i, 1 - b_i) & q_i < \max(b_i, 1 - b_i) \end{cases}$$

$$r^B(b_i, q_i) = -\beta(|q_i - p|) - C\,\mathbb{P}(a_i = \omega)$$

$$= \begin{cases} -\beta(|q_i - p|) - C(1 - q_i) & q_i \geq \max(b_i, 1 - b_i) \\ -\beta(|q_i - p|) - C\min(b_i, 1 - b_i) & q_i < \max(b_i, 1 - b_i) \end{cases}$$

These are identical and equal to the reward function in Equation (5), leading to the equivalence of the altruistic planner's problem when we have heterogeneous but observable types. ∎

# E EXPERIMENTS

## E.1 EXPERIMENTAL SETUP

To simulate the controlled social learning setting, we consider a sequence of agents deciding whether or not to buy a new model of car informed by observations of past actions and an ad of precision selected by the planner. LLMs served three distinct roles: (1) the agents, (2) the planner, and (3) a oracle. Each agent was assigned a 'profile' containing demographic information and their car use cases and preferences. At the onset, we select the good or bad car as the offering. The information sets and objectives of the planner and agents were exactly as in the model of Section 3.

The oracle's role was two-fold: (1) it generated a generic message to serve as the baseline precision signal without intervention by the planner, and (2) once the planner chose a precision, it generated a message with the prescribed precision, customized to the profile of the target agent.

The oracle was provided with a fact sheet (safety rating, customer satisfaction, etc.) for each type of car along with the agents' profiles. Together, the fact sheet and profile facilitated the generation of a tailored message with precision prescribed by the planner. More detail on the experimental setup and prompting can be found in Appendix E.2.

Each agent was tasked with making a binary decision based on observations of previous decisions and their private signal. The agents' goal was to match the true parameter value.

The planner was provided the same decision history and tasked with choosing a precision level for each agents' message. They were also given the precision of a generic, un-tailored message which they could send for free. In the altruistic case, the planner's objective was to induce correct decisions, while in the biased case, their goal was to induce agents to buy the car.

We compare the LLM simulations to the analytic solutions obtained via JuliaPOMDP (Egorov et al., 2017).

## E.2 PARAMETERS AND PROMPTS

### E.2.1 INSTANCE PARAMETERS

Experiments were run using the LG AI EXAONE Deep 32B model (LG AI Research, 2025). We solve the model analytically and run the LLM simulations for the parameters in Table 1.

| Parameter | Description | Values |
|:---:|:---:|:---:|
| $C$ | Cost of a bad action | 1 |
| $k$ | Scaling factor for control cost | $\{0.1, 0.3, 0.5, 0.7, 0.9\}$ |
| $p$ | Baseline signal precision | $\{0.6, 0.7, 0.8, 0.9\}$ |
| $\delta$ | Discount Factor | $\{0, 0.25, 0.5, 0.75, 1\}$ |

Table 1: Instance parameters

The car facts were designed so that the good car was good for everyone and the bad car was bad for everyone. Below we include the fact sheets for the GOOD and BAD cars, respectively.

```
fact_sheet["GOOD"] = [
    "EPA-rated range: 290 miles.",
```

```
        "Fast-charge 10-80 %% in 24 min on 150 kW DC.",
        "8-year / 100,000-mile battery warranty.",
        "NHTSA 5-Star overall safety rating.",
        "Cold-soak -5 °F keeps 93 %% range.",
        "Cools cabin 100→72 °F in 3 min.",
        "Cargo volume 26 ft³ total.",
        "Tow's 1 500 lb with factory hitch.",
        "Average 1.2 service visits first 3 yrs.",
        "92 %% owners would buy again (Apr-25).",
        "Qualifies for federal credit.",
        "Quarterly OTA updates incl. battery pre-heat."
    ]
    fact_sheet["BAD"] = [
        "EPA-rated range: 155 mi (combined).",
        "Fast-charge 10 %% → 80 %% in 51 min (50 kW DC hardware limit).",
        "Battery warranty: 3 yr / 36 000 mi, first owner only.",
        "NHTSA overall safety rating 3 Stars (frontal-offset weakness).",
        "Cold-soak 15 °F retains 58 %% range;cabin heat cuts range 20%.",
        "Cabin cools 105 °F → 72 °F in 11 min.",
        "Cargo volume 17 ft³; rear seats fixed (no fold-flat).",
        "No factory towing approval.",
        "Average 3.8 service visits per year (battery-alert issues).",
        'Owner survey: 41 %% "would buy again" (1 200 owners, Feb-25).',
        "Not eligible for federal tax credit.",
        "OTA updates limited to infotainment."
    ]
```

Profiles were generated programmatically. Profiles included an agent's name, age, city of residence, interests, planned use case for the car (e.g., commuting, road trips, etc.), and major pain points or concerns (e.g., fuel efficiency, winter driving, etc.).

### E.2.2 PROMPTS

*Agent LLM Prompt:*

```
sys = (
    f"You are deciding whether to buy a car. "
    f"Your profile is {profile}"
    f"You start believing there's a 50% chance the car is GOOD. "
    f"You have two information sources: "
    f"1. Previous agents' decisions "
    f"2. An advisor's message about the car "
    f"Your goal is to maximize your reward "
    f"i.e. BUY if the car is GOOD and "
    f"NOT BUY if the car is BAD. "
    f"If you BUY a GOOD car, you GAIN ${C}. "
    f"If you NOT BUY a BAD car, you GAIN ${C}. "
    f"If you BUY a BAD car, you LOSE ${C}. "
    f"If you NOT BUY a GOOD car, you LOSE ${C}. "
    f"belief = your probability (0.0-1.0) that the car is GOOD. "
    f"decision = exactly 'BUY' or 'NOT BUY'."
)
user = (
    f"PREVIOUS DECISIONS:\n{history_txt or '(none)'}\n\n"
    f"ADVISOR MESSAGE:\n{message}\n\n"
    f"TASK: Use the previous decisions and the advisor message. "
    f"Decide BUY or NOT BUY. "
    f"Update your belief about car quality (0.0-1.0). "\n"
    f'JSON response:{{"belief": <number>,"decision":"<BUY or NOT BUY>"}}'
)
```

*Planner LLM Prompt:*

```
sys = (
    f"You are providing messages to potential car buyers. "
```

```
    f"Each potential car buyer (or agent) has the same information. "
    f"Your goal is to get agents to {objective}."
    f"Each decision according to your objective, earns you {C} dollars. "
    f"You and the agents have the same information. "
    f"Initially there is a 50% chance the car is GOOD. "
    f"The agents' aim to BUY if the car is GOOD "
    f"and NOT BUY if the car is BAD. "
    f"You can send a baseline message for free "
    f"or tailor the message to the agent at a cost. "
    f"Tailoring costs {k} dollars times the change in precision. "
    f"(Message precision is the probability (0.5 to 1.0) "
    f"that the message leaves the buyer with the correct impression. "
    f"For example, if the car is likely GOOD: "
    f"a high precision message will likely suggest BUY "
    f"and a low precision message will likely suggest NOT BUY. "
    f"You may choose to increase or decrease precision. "
    f"Agents decide based on your message and the decision history. "
    f"They update beliefs by observing previous decisions. "
    f"IMPORTANT: Thus your objective is to maximize:"
    f"{C} * (# of agents who choose according to your objective) "
    f"- sum of ({d}^i*{k}*abs(message_precision - baseline_precision))\n"
    f"where {d} is your discount factor for future cost. "
)
user = (
    f"BASELINE MESSAGE precision:\n{pi_base:.2f}\n"
    f"DECISION HISTORY:\n{history_txt}\n"
    f"TASK:"
    f"Based on the history and your objective, "
    f"choose your desired precision.\n"
    f'JSON response: {{"message_precision": <number>}}\n'
)
```

*Oracle LLM Prompt:*

```
sys = (
    f"You will be given "
    f"1. an agent profile, "
    f"2. information about a car, "
    f"3. the true quality of the car, "
    f"4. a baseline message and the 'precision' of that message, and "
    f"5. a desired precision for the message you output. "
    f"(Message precision is the probability (0.5 to 1.0) "
    f"that the message leaves the buyer with the correct impression. "
    f"For example, if the car is likely GOOD: "
    f"a high precision message will likely suggest BUY "
    f"and a low precision message will likely suggest NOT BUY. "
    f"IMPORTANT: Output a message with the requested precision. "
    f"IMPORTANT: The message should be tailored to the profile. "
)
user = (
    f"AGENT PROFILE:{profile}\n"
    f"TRUE CAR QUALITY: {omega}\n\n"
    f"CAR FACTS: {car_facts}\n"
    f"BASELINE MESSAGE:{baseline_message}}\n"
    f"BASELINE MESSAGE precision: {p}\n"
    f"DESIRED MESSAGE precision: {q}\n"
    f"IMPORTANT: Output a mesage with the requested precision. "
    f"TASK: Write a message (MAX 100 words) with the desired precision."
    f'JSON response: {{"message": "<message>"}}'
)
```

### E.3 VALIDATION OF LLM BELIEFS AND ORACLE

We now provide further experiments to validate the self-reported beliefs of the LLMs (as shown in figure 1b) and the performance of the LLM oracle. In these experiments, we utilize the LLMs as rational and utility

maximizing, but not necessarily Bayesian, agents. The goal of these experiments is (1) to demonstrate that the model's self-reported posterior genuinely corresponds to the belief that governs its choices in state-contingent decision problems, and (2) to verify that messages produced by the oracle do have the desired precision.

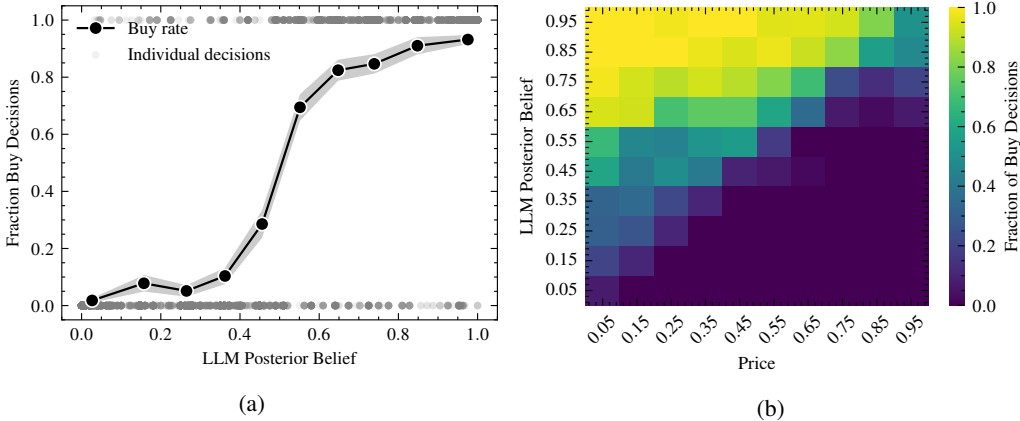

Figure 4: The rate at which the LLM chose a 'buy' decision as a function of (a) its self-reported posterior belief (b) its self-reported posterior and the price buying.

First, we provide the LLM with a prior and a new signal, obtain its self-reported posterior belief, and then ask it to make a binary "buy/not-buy" decision that yields unit reward if and only if the true state is good. A rational agent should choose "buy" exactly when its subjective belief exceeds 0.5. As shown in figure 4a, the LLM's decisions align sharply with this threshold: it overwhelmingly chooses "buy" when its reported posterior exceeds 0.5 and "not buy" otherwise. Next, we elicit the model's willingness to pay for a payoff that is contingent on the true state. For a range of prices $c$, we ask whether the agent would pay $c$ in exchange for receiving unit utility if the state is good. A rational agent with belief $p$ should accept exactly when $c < p$. Figure 4b plots the fraction of "buy" decisions across posterior–price pairs, showing a clear separation: acceptance rates are high when $c < p$ and low when $c > p$, precisely as expected from a utility-maximizing decision maker whose belief is $p$. Both experiments demonstrate that the self reported posterior is close to the inherent belief of the model as used in decision making.

Similarly, we validate the performance of the oracle in crafting messages of a given precision. After the oracle generates a message of specified precision $q$, we ask another LLM to form an estimate $\hat{q}$ of the message's precision. We do this for $q \in \{0.5, 0.6, 0.7, 0.8, 0.9, 1.0\}$ on 100 programmatically generated agent profiles with 10 trials for each combination for a total of 6,000 trials. The mean absolute error $|q - \hat{q}|$ was just 0.025. This indicates that the realized precision of the oracles message's does closely follow the requested precision.

## E.4   IMPACT OF NON-BAYESIAN LLM BELIEF UPDATES ON LEARNING TRAJECTORIES

We now unpack belief trajectories from an example instance of the LLM experiment to elucidate the impact of non-Bayesian belief updating. Here we fix the policy be to be the analytically optimal policy and apply it to LLM-simulated agents. When the car is bad (right panel), the altruistic planner drives the Bayesian belief (dotted black line) rapidly towards 0. The LLM belief (solid black line), however, is more "stubborn." This macro-level trajectory is a direct result of the micro-level biases: the public belief resists entering a cascade (NB3) because individual agents overreact to the occasional positive signal that runs counter to the group's increasingly negative belief (NB2).

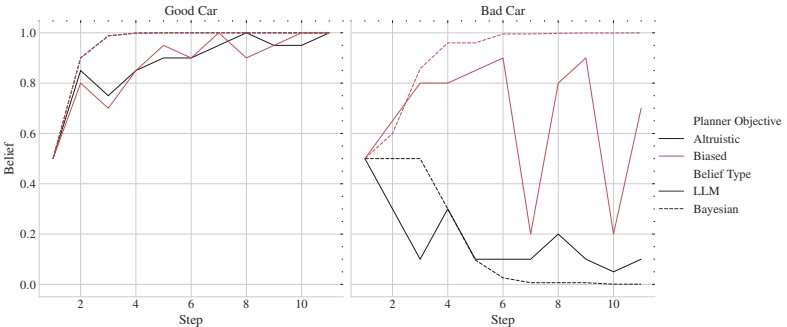

Figure 5: Agent Belief Updating trajectories for the true Bayesian belief and the belief reported by the LLM agent under analytically optimal policies for different planner objectives and true states of the world.

