# OpenReview forum: "Steering the Herd: A Framework for LLM-based Control of Social Learning"
_ICLR.cc/2026/Conference — ICLR 2026 Oral_

### Official Review · Reviewer_v6hQ · 2025-10-30

**Soundness:** 4
**Presentation:** 4
**Contribution:** 4
**Rating:** 8
**Confidence:** 3

**Summary:**

The model introduces a binary persuasion problem of a long-lived designer interacting with a sequence of short-lived agents that learn from the prior action sequence. This problem is related to social learning among LLM agents as well as human societies. They prove tractability of the model with Bayesian agents and investigate non-Bayesian LLM agents in simulations.

**Strengths:**

The paper thoroughly sets up its problem, and is very well-written. I found the integration of experiments alongside a (mostly) theory paper convincing, as the authors stated departures from their Bayesian assumption in the beginning of their experiments section.

**Weaknesses:**

The paper considers and names several restrictions in particular on the information structure and short-livedness of agents, but does not say something on how these affect outcomes. While I understand that for the tractability of the theoretical model all of them are essential, the fact that the authors run experiments would allow for an investigation of what other dynamics are possible with variations on, e.g., the stringent public information assumptions.

**Questions:**

- Does your setting admit a revelation principle?
- (LLM-)Experimentally, which assumptions lead to significantly different dynamics compared to your current simulation exercise?

---

> ### Author Response · Authors · 2025-11-22
> **Author Response Part I**
>
> We are pleased to hear that the reviewer found our combination of theoretical and experimental analysis convincing. We are also appreciative of their feedback which broaches several important points addressed below:
>
> ## Model Assumptions
>
> The reviewer makes a great point \-- it is worthwhile to discuss the ways in which we might strip away the assumptions of our model. Below we discuss some generalizations of our problem and their feasibility. We have also incorporated these generalizations in our conclusion.
>
> ### Beyond the Binary Case
>
> We consider a binary setting as in classical works in sequential social learning (e.g., [1]). This is primarily for ease of exposition. As in [2], it is possible to relax this to a finite setting, but there is "significant algebraic cost". For $k$ states each with their own respective optimal action, the belief would be a point on the $k-1$-dimensional simplex and each action would be optimal within a convex region of the simplex.
>
> ### More General Signal Structures
>
> The most significant challenge in considering more general signal structures is ensuring that the convexity result of Theorem 2 still holds. We currently prove the result for the binary and symmetric case where both the state (public belief) and action (chosen precision) are scalar. It is not clear whether the neat threshold structure of our results (e.g., the simple form of the optimal altruistic policy in Theorem 3\) would still hold in higher dimensional settings. Finding the most general class of signal structures for which such a result holds is an interesting problem for future study.
>
> ### More General Loss Functions
>
> With discrete actions, any expected utility maximizing agent's decision reduces to choosing convex decision regions over the belief simplex. A change in loss functions then amounts to shifting the corresponding decision boundaries. In the binary case, the 0-1 loss function we assume has a decision threshold at 0.5, i.e., an agent's optimal choice is $G$ if their posterior belief is above 0.5. For a different loss function this specific threshold may change; however, so long as agent preferences are known to all, future agents and the planner will still be able to properly infer information from actions and maintain the public belief. Thus, the thresholds and boundaries of our analysis may shift, but we do not expect a change in the qualitative results.  In a related generalization, we have extended our results for the altruistic planner to the case of two agent types with opposing preferences (i.e., loss functions) in Appendix D.
>
> ### Non-Myopic Agents
>
> There are some works that study sequential social learning where agents are not myopic (e.g., [3]). Here, the agents' optimal strategies can become much more complex, making equilibrium analysis more challenging.
>
> ## Experimentation Under Relaxed Assumptions
>
> Experiments can indeed be conducted under relaxed assumptions; however, extending the theory to such settings is a significant undertaking. Until theoretical results are obtained under relaxed assumptions, experimental results would lack a meaningful analytical benchmark for comparison. Hence, we defer such experiments to future work.
>
> In particular, consider relaxing the public information assumption mentioned by the reviewer to allow for private agent preferences or private signal precision. In this case, the agents and planner cannot perfectly infer the private information from observable actions as they can in case 1 of Eqn. 2 in our work. Thus, there is not a common understanding of the public belief. Empirical work has shown such changes can lead to significant changes in learning dynamics [4].
>
> As such, and given our focus on analytical results, we defer more general experimentation settings to future work.
>
> ## References
>
> [1]	S. Bikhchandani, D. Hirshleifer, and I. Welch, “A Theory of Fads, Fashion, Custom, and Cultural Change as Informational Cascades,” https://doi.org/10.1086/261849, vol. 100, no. 5, pp. 992–1026, 1992, doi: 10.1086/261849.
>
> [2]	L. Smith and P. Sørensen, “Pathological outcomes of observational learning,” Econometrica, vol. 68, no. 2, pp. 371–398, 2000, doi: 10.1111/1468-0262.00113.
>
> [3]	I. Bistritz, N. Heydaribeni, and A. Anastasopoulos, “Informational Cascades With Nonmyopic Agents,” IEEE Trans. Autom. Control, vol. 67, no. 9, pp. 4451–4466, Sept. 2022, doi: 10.1109/TAC.2022.3165483.
>
> [4]	T. Gagnon-Bartsch and B. Bushong, “Heterogeneous Tastes and Social (Mis)Learning”.

---

> ### Author Response · Authors · 2025-11-22
> **Author Response Part II**
>
> ## Revelation principle
>
> The reviewer’s question on a revelation principle is important and makes a key distinction between our work and the literature on mechanism design. Revelation principles are primarily of import when there is the potential for information asymmetry and the designer seeks to elicit the agents' private information. For example, in auctions, the auctioneer may choose a mechanism where bidders report their values for the items for sale, and the revelation principle allows us to restrict attention to incentive-compatible mechanisms where bidders report truthfully. In our setting, this information asymmetry in the sense of private valuations of the agents is not present. Also, we are explicitly not allowing our social planner to open up a communication channel for the agents to report their signal realizations or other information. We instead work in a more constrained setting where the social planner only observes agents' binary choices and can only intervene by changing the precisions of the agent's signals, without being able to elicit agents' signal realizations or exact beliefs through a reporting system. We think this is the more realistic assumption in our applications, where citizens typically do not directly report their impressions about a technology or new product to the government.

---

### Official Review · Reviewer_gvhy · 2025-10-31

**Soundness:** 2
**Presentation:** 3
**Contribution:** 1
**Rating:** 2
**Confidence:** 4

**Summary:**

This paper introduces a formal framework for controlled social learning where a centralized planner (e.g., an LLM-based mediator) chooses the precision of private signals sent to sequential agents. The planner can be altruistic (maximizing social welfare) or biased (steering agents toward a preferred action), and faces costs for altering signal precision. The authors characterize optimal planner policies as functions of evolving public belief, prove structural properties (e.g., convexity of the altruistic value function), and show how informational externalities shape long-run outcomes. LLM-based simulations validate the theory, revealing that adaptive planners can substantially influence collective beliefs and welfare and that modern LLMs can exhibit emergent strategic behaviors—highlighting both opportunities and risks of algorithmic information mediation.

**Strengths:**

- The understanding and introduction of Bayesian persuasion (information design) are accurate.
- The entire article is presented well, with a clear structure and easy to understand.

**Weaknesses:**

**1**

Currently, this manuscript appears to lack sufficient literature review, which makes the position and contribution problematic.

Contribution 1 is not as claimed. The authors claim:
> We introduce the first formal model that integrates a dynamic control problem for a centralized information planner with the mechanism of sequential social learning.

There is a series of articles that the authors have not considered: the combination of Bayesian persuasion (information design) and RL. In fact, in the past 5 years, there have been a large number of variants such as online persuasion and multi-receiver variants, etc. (while the authors' citations on information design are the latest only up to 2000).

Talking about RL is not off-topic—considering that the authors position their article as a control problem (dynamic programming). Then this series of articles must at least be discussed.

Farsighted agents is also a rapidly developing topic in the persuasion community. The social learning mentioned in this manuscript is indeed a novel point, but it is also a common problem in the multi-agent RL field. Therefore, the claim of contribution 1 is incorrect.

In addition, the citations on LLM on information design / Bayesian persuasion are not comprehensive enough (only focusing on the work of two or three labs), and the discussion is too limited.

**2**

This manuscript is submitted to the Primary Area: alignment, fairness, safety, privacy, and societal considerations.

However, this manuscript is about a stylized dynamic programming problem with a lot of assumptions. According to the authors' position, it seems that the entire related part should be in Section 6: EVALUATION VIA LLM-BASED SIMULATION.

The steps in section 6 by the authors are: “analyze the behavior of LLM agents to identify key deviations from Bayesian rationality”. The remaining logic seems to be: LLM is non-Bayesian, but LLM can still exhibit results that do not deviate much from the theoretical analysis results of Bayesian-rational agents.

Regarding the player setting, there are 2 gaps:

- non-Bayesian players vs human; non-Bayesian is a very rough concept, as long as it does not perfectly conform to Bayesian rationality, it counts. This is just one trait of humans. Not satisfying Bayesian rationality is not sufficient to conclude that it is close to humans. Humans have many other traits, and from an agent-based perspective, they can also be endowed with different personalities.
- LLM vs human; this point is currently very controversial in the literature.

Therefore, the authors' experiments may only illustrate that this algorithm has a certain robustness and can have good effects even when agents do not satisfy Bayesian-rationality. But this cannot indicate that this algorithm is more realistic, nor can it be said to be an effect brought by LLM.

In addition, the main text does not provide the LLM's role-play settings, and the experiments do not involve human participation or human data. Therefore, I believe the experiments and their claims are mismatched.

**3**

The abstract must be changed to a single paragraph.

**Questions:**

N/A

---

> ### Author Response · Authors · 2025-11-22
> **Author Response Part I**
>
> We thank the reviewer for their detailed and constructive feedback. We have applied it to better situate our work within a broader context and more precisely delineate the conclusions drawn from our LLM experiments as described below. We hope the following comments, revisions, and additional results address the reviewer's concerns:
>
> ## Related Work and Novelty
>
> Owing to the reviewers feedback, we have significantly expanded the discussion of related work in our paper (Section 2), explicitly distinguishing our setting from those discussed in their review. We will first address this at a high level before more thoroughly considering each field.
>
> To understand the novelty of our work and its relation to other literature, we begin with our motivation: the ability of information mediators to control society by shaping the information available to people. Specifically, do such planners have the ability to personalize private messages to induce herding behavior on actions which may or may not be socially favorable? This problem is particularly relevant in the context of LLMs as they have been shown to be persuasive and have the ability to personalize content at scale.
>
> To study this, a model requires (1) an information designing planner, (2) agents who learn from each other and the information designed by the planner over time, and (3) a mechanism for learning that can capture herding phenomena. Literature in Bayesian persuasion (BP) + and reinforcement learning (RL) considers information designers but generally does not capture (2) nor (3). Multi-agent RL (MARL) works study (2) but often lack (1) and/or (3). Finally, the work in LLMs and information design consider (1) but lack (2) and (3). Below, we more thoroughly address the most salient works of each line of work in our context.

---

> ### Author Response · Authors · 2025-11-22
> **Author Response Part II**
>
> ### BP+RL
>
> As the reviewer notes, there is growing recent work at the interface of online learning and Bayesian persuasion/information design. These works study online sequential persuasion under unknown environment parameters, in which a sender interacts repeatedly with short-lived receivers while lacking knowledge of the prior over states and/or receivers’ utility functions (e.g., [1],[2],[3],[4]). A related set of papers examines sequential persuasion in dynamic environments or MDPs, introducing state transitions and/or foresight by the receiver; these models emphasize history-dependent or promise-based signaling policies and often connect to reinforcement learning (e.g., [5],[6],[7]). Across these works, the methodological focus lies in learning-based performance guarantees such as sublinear regret relative to the best signaling scheme in hindsight and/or approximation algorithms under computational constraints.
>
> In contrast, our model does not require the planner to learn unknown parameters; sequential dependence arises from social learning among receivers, rather than from the sender updating beliefs about the environment. Moreover, the planner is not informationally advantaged: it does not observe the underlying state or private signals and does not acquire information from interactions. Within this setting, we are able to characterize optimal policies for the planner. While sequential, our setting is not an online-learning or RL environment: actions do not modify the underlying payoff-relevant state (no reward externalities), and the planner does not learn unknown environmental parameters. The planner updates its knowledge only through observing agents’ actions, which reflect social learning and belief propagation rather than payoff exploration or state evolution. Sequential dependence therefore arises from information externalities, as agents' actions shape the beliefs of future agents.
>
> ### MARL
>
> In the MARL literature, social learning often refers to settings where agents directly influence each other's rewards whether through reward sharing, a cooperative game, or some other mechanism (e.g., [8],[9],[10],[11]). In contrast, our social learning model focuses on sequential (observational) social learning via *information externality* (observational inference regarding an underlying state) rather than *reward externality*. Specifically, our model of social learning captures herding phenomena while those in the MARL literature generally do not.
>
> We specifically address [12] and [13], which share closer thematic ties to our work regarding observational dynamics and influence. [13] studies how social learning emerges as a policy in decentralized agents (i.e., agents learning to observe and value the actions of experts). In [12], an agent learns by observing the actions of a teacher agent. Both works primarily focus on the agent's problem of learning through observation whereas we focus instead on the planner's control problem when social learning is assumed to be in effect.
>
> ### LLMs and Information Design
>
> Finally, we have expanded our references to include recent works on LLMs and information design ([14],[15],[16]). [14] considers one-shot interactions and uses a BP framework to measure LLM persuasiveness. [15] studies a two-player, repeated persuasion game and designs a Monte Carlo Tree Search algorithm to play the role of the sender. [16] applies findings from online BP (e.g., [17]) to a setting where a sender LLM persuades a receiver LLM with the goal of ensuring alignment of the receiver. All three consider one-on-one interactions and, thus, do not capture the social dynamics we aim to model.

---

> ### Author Response · Authors · 2025-11-22
> **Author Response Part III**
>
> ## Experimental Claims
>
> We thank the reviewer for the care taken in considering the validity of our experimental results and conclusions. We respond to the reviewer's comments below and have also edited Section 6 to clarify our exact claims and revise the presentation. We hope these responses address the reviewer's concerns.
>
> First, we wish to clarify the intended logic of section 6\. Here our goal is not to prove that LLMs are good simulators of human behavior. Instead, our focus is on the *planner LLM*, and we aim to demonstrate the following:
>
> 1. **The importance and impact of social learning control:** Planners (both analytical and LLM) can dramatically improve their own utility by accounting for and capitalizing on social learning dynamics. This may help or harm social welfare contingent upon the alignment between planner and agent objectives.
> 2. **The robustness of our analytical characterization:** LLM planners choose policies surprisingly similar to the non-obvious analytically optimal policies despite facing non-Bayesian agents in other LLMs. This shows our analytical characterization is, to some degree, robust against such settings, lending credence to the utility of our framework despite the assumptions made on the planner noted in Remark 2\.
> 3. **The emergent strategic behavior of LLM:** LLM planners exhibit sophisticated emergent strategic behavior in their choice of policies. Not only are LLM policies close to the optimal policies, they also seem to deviate in ways which account for the non-Bayesian nature of the LLM agents they face. This demonstrates a surprising and potentially dangerous capability, motivating further study of ways to mitigate the risk of LLMs as they are increasingly prevalent in the role of information mediator.
>
> (1) and (2) serve to substantiate the importance, validity, and impact of the analysis contained in previous sections.
>
> ### Non-Bayesian vs. LLM vs. Human
>
> We agree that "Non-Bayesian" is a broad concept. Our results shows something stronger: not only are the LLM agents non-Bayesian, but they exhibit a cognitive bias in their belief updating (Fig 1b and claims (NB1) and (NB2)):
>
> 1. They overreact to surprising information (signals counter to their prior).
> 2. They underreact to expected information.
>
> This specific behavior has been observed empirically in human studies (e.g., [18] and [19]). While this does not definitively prove that LLMs are good approximators of humans, it does provide stronger evidence than non-Bayesian updates on their own. We agree that incorporating human data directly into our framework would be the gold standard for making stronger claims about the realism of the model we consider. Collecting and working with such data is an interesting line of inquiry for future work.
> There is also increasing interest in settings where both parties (planner and agent) are LLMs. In this case, it is not necessary that LLMs simulate human behavior. It is of interest to study emergent behavior in such settings as in (3) above.
> In addition, we have also augmented the beginning of Section 6 with greater detail regarding the experimental setup and prompting (drawn from Appendix E). We believe this improves the clarity of the experiments and thank the reviewer for the suggestion.
>
> ## Abstract Formatting
>
> We agree with the reviewer and have reformatted the abstract accordingly.

---

> ### Author Response · Authors · 2025-11-22
> **Author Response Part IV (References)**
>
> ## References
> [1]	M. Castiglioni, A. Celli, A. Marchesi, and N. Gatti, “Online Bayesian Persuasion,” in Advances in Neural Information Processing Systems, Curran Associates, Inc., 2020, pp. 16188–16198. Accessed: July 30, 2025. [Online]. Available: https://proceedings.neurips.cc/paper/2020/hash/ba5451d3c91a0f982f103cdbe249bc78-Abstract.html
>
> [2]	F. Bacchiocchi, M. Bollini, M. Castiglioni, A. Marchesi, and N. Gatti, “Online Bayesian Persuasion Without a Clue,” Adv. Neural Inf. Process. Syst., vol. 37, pp. 76404–76449, Dec. 2024, doi: 10.52202/079017-2434.
>
> [3]	M. Castiglioni, A. Marchesi, A. Celli, and N. Gatti, “Multi-Receiver Online Bayesian Persuasion,” in Proceedings of the 38th International Conference on Machine Learning, PMLR, July 2021, pp. 1314–1323. Accessed: Nov. 15, 2025. [Online]. Available: https://proceedings.mlr.press/v139/castiglioni21a.html
>
> [4]	S. Agrawal, Y. Feng, and W. Tang, “Dynamic Pricing and Learning with Bayesian Persuasion,” Adv. Neural Inf. Process. Syst., vol. 36, pp. 59273–59285, Dec. 2023.
>
> [5]	J. Wu et al., “Sequential Information Design: Markov Persuasion Process and Its Efficient Reinforcement Learning,” Feb. 22, 2022, arXiv: arXiv:2202.10678. doi: 10.48550/arXiv.2202.10678.
>
> [6]	S.-T. Su, V. G. Subramanian, and G. Schoenebeck, “Bayesian Persuasion in Sequential Trials,” in Web and Internet Economics, M. Feldman, H. Fu, and I. Talgam-Cohen, Eds., Cham: Springer International Publishing, 2022, pp. 22–40. doi: 10.1007/978-3-030-94676-0_2.
>
> [7]	M. Bernasconi, M. Castiglioni, A. Marchesi, and M. Mutti, “Persuading Farsighted Receivers in MDPs: the Power of Honesty,” Adv. Neural Inf. Process. Syst., vol. 36, pp. 14987–15014, Dec. 2023.
>
> [8]	P. Chelarescu, “Deception in Social Learning: A Multi-Agent Reinforcement Learning Perspective,” June 09, 2021, arXiv: arXiv:2106.05402. doi: 10.48550/arXiv.2106.05402.
>
> [9]	J. Oh and S. F. Smith, “A few good agents: multi-agent social learning,” in Proceedings of the 7th international joint conference on Autonomous agents and multiagent systems - Volume 1, in AAMAS ’08. Richland, SC: International Foundation for Autonomous Agents and Multiagent Systems, May 2008, pp. 339–346.
>
> [10]	J. Hao, H.-F. Leung, and Z. Ming, “Multiagent Reinforcement Social Learning toward Coordination in Cooperative Multiagent Systems,” ACM Trans Auton Adapt Syst, vol. 9, no. 4, p. 20:1-20:20, Dec. 2014, doi: 10.1145/2644819.
>
> [11]	J. Z. Leibo, V. Zambaldi, M. Lanctot, J. Marecki, and T. Graepel, “Multi-agent Reinforcement Learning in Sequential Social Dilemmas,” Feb. 10, 2017, arXiv: arXiv:1702.03037. doi: 10.48550/arXiv.1702.03037.
>
> [12]	D. Borsa, B. Piot, R. Munos, and O. Pietquin, “Observational Learning by Reinforcement Learning,” June 20, 2017, arXiv: arXiv:1706.06617. doi: 10.48550/arXiv.1706.06617.
>
> [13]	K. K. Ndousse, D. Eck, S. Levine, and N. Jaques, “Emergent Social Learning via Multi-agent Reinforcement Learning,” in Proceedings of the 38th International Conference on Machine Learning, PMLR, July 2021, pp. 7991–8004. Accessed: Nov. 18, 2025. [Online]. Available: https://proceedings.mlr.press/v139/ndousse21a.html
>
> [14]	Z. Cheng and J. You, “Towards Strategic Persuasion with Language Models,” Sept. 26, 2025, arXiv: arXiv:2509.22989. doi: 10.48550/arXiv.2509.22989.
>
> [15]	M. Raifer, G. Rotman, R. Apel, M. Tennenholtz, and R. Reichart, “Designing an Automatic Agent for Repeated Language–based Persuasion Games,” Trans. Assoc. Comput. Linguist., vol. 10, pp. 307–324, Mar. 2022, doi: 10.1162/tacl_a_00462.
>
> [16]	F. Bai et al., “Efficient Model-agnostic Alignment via Bayesian Persuasion,” May 29, 2024, arXiv: arXiv:2405.18718. doi: 10.48550/arXiv.2405.18718.
>
> [17]	M. Bernasconi, M. Castiglioni, A. Celli, A. Marchesi, F. Trovò, and N. Gatti, “Optimal Rates and Efficient Algorithms for Online Bayesian Persuasion,” in Proceedings of the 40th International Conference on Machine Learning, PMLR, July 2023, pp. 2164–2183. Accessed: Nov. 15, 2025. [Online]. Available: https://proceedings.mlr.press/v202/bernasconi23a.html
>
> [18]	C. Ba, J. A. Bohren, and A. Imas, “Over- and Underreaction to Information,” SSRN Electron. J., 2022, doi: 10.2139/ssrn.4274617.
>
> [19]	K. Chan, G. Charness, C. Dave, and J. L. Reddinger, “On Prior Confidence and Belief Updating,” May 14, 2025, arXiv: arXiv:2412.10662. doi: 10.48550/arXiv.2412.10662.

---

> > ### Comment · Reviewer_gvhy · 2025-11-27
> >
> > I acknowledge that the authors have addressed these issues:
> >
> > 1. Previously, the authors' literature review on persuasion only covered works up to 2020 and ignored important literature on sequential persuasion (omissions that made the contribution claims problematic). The authors have now indeed added relevant discussion.
> > 2. There were no prompt settings provided previously; the authors have now provided them.
> > 3. The abstract was previously formatted as two paragraphs; it has now been corrected to a single paragraph.
> >
> > Based on the factors above, I consider the authors' revisions and rebuttal to be satisfactory. Therefore, I am raising my score from 2 to 4.
> >
> > However, I still disagree with the following points in the authors' rebuttal:
> >
> > **1**
> >
> > The literature review remains insufficiently comprehensive. I suggest the authors leverage state-of-the-art AI tools to conduct literature surveys in domains they are less familiar with.
> >
> > The argument regarding the MARL section is incorrect: "Multi-agent RL (MARL) works study (2) but often lack (1) and/or (3)." where " (1) an information designing planner, (2) agents who learn from each other and the information designed by the planner over time, and (3) a mechanism for learning that can capture herding phenomena."
> >
> > 1. The claim "Specifically, our model of social learning captures herding phenomena while those in the MARL literature generally do not" is incorrect. MARL does involve (1) + (3). Recommended references:
> >     1. Artificial intelligence meets minority game: toward optimal resource allocation
> >     2. Mesoscale effects of trader learning behaviors in financial markets: A multi-agent reinforcement learning study
> >     3. OASIS: Open Agents Social Interaction Simulations on One Million Agents
> >     4. Social Amplification Can Help Solve the Credit Assignment Problem in Collective Learning
> > 2. Papers combining information design and MARL were overlooked. MARL does involve (1) + (2). Recommended references:
> >     1. Information Design in Multi-Agent Reinforcement Learning
> >
> > P.S.: "(3) a mechanism for learning that can capture herding phenomena" is a very specific niche. While it is true that few papers satisfy (1) + (2) + (3) simultaneously, imposing this restriction also leads the authors to narrow the generality and significance of the contribution.
> >
> > **2**
> >
> > Regarding Non-Bayesian vs. LLM vs. Human, I believe there are certain gaps that the authors can neither avoid nor resolve. I view this as a limitation rather than a weakness. It is acceptable as long as the authors discuss these gaps in detail within the paper.
> >
> > Recommended reference:
> > - Take caution in using LLMs as human surrogates

---

> > > ### Comment · Reviewer_gvhy · 2025-11-27
> > >
> > > Corrections to my previous response:
> > >
> > > - A typo: The herding effect in MARL involves (2) + (3), not (1) + (3).
> > >
> > > - Although the paper "OASIS: Open Agents Social Interaction Simulations on One Million Agents" is not RL (I mistakenly categorized it under MARL in my previous response), it combines LLMs and social learning, and explicitly considers the herding effect.

---

> > > > ### Author Response · Authors · 2025-12-03
> > > >
> > > > We thank the reviewer for increasing their rating and for their feedback and engagement. Below we address the reviewer's response. We hope that these clarifications and corresponding revisions to our paper tackle any remaining concerns.
> > > >
> > > > ## Literature Positioning and Novelty
> > > >
> > > > We appreciate the reviewer pointing us to further works in the literature (\[2\]-\[6\]) that address elements of “social learning” and/or notions of “herding” using multiagent learning frameworks. Below, we elaborate on how those works relate to and differ from our problem of interest.
> > > >
> > > > Our motivation is to study the optimal policies of a planner (LLM) controlling the information ecosystems of connected agents (users). In these systems, agents make decisions based on both private signals curated by the planner and coarse observational inferences about previous users’ signals, potentially leading to information cascades (herding) where social learning breaks down and private information is lost. These phenomena are observed both theoretically and empirically (see \[1\] and references therein). To our knowledge, ours is the first work introducing an information designer to this setting and studying the dynamic planner’s problem therein. This allows us to answer important questions such as
> > > >
> > > > * How might an altruistic planner mitigate or manage the social welfare risk of herding? Does herding still occur under such a planner?
> > > > * Could a biased planner use such breakdowns in social learning to their own benefit? If so, how does this affect social welfare?
> > > >
> > > > It is worthwhile to mention that the study of such problems requires a planner (the information mediator). This is missing in \[2\]-\[5\] mentioned by the reviewer (more details below). \[6\] includes an information designer, but considers only one agent (receiver). Thus, their model cannot capture the inter-agent learning central to social learning and herding.
> > > >
> > > > Another fundamental distinction between our work and the MARL literature lies in the mechanism of social learning. In MARL, "social learning" often refers to settings where agents directly influence each other's rewards—whether through reward sharing, cooperative games, or resource competition. In contrast, our model focuses on sequential social learning via *information externalities* rather than *reward externalities*. In our setting, agents do not modify the underlying payoff-relevant state; rather, their actions shape the beliefs of future agents via observational inference regarding an underlying state. Information externalities are naturally suited to studying the control of opinion/belief formation which is the central concern of our work. Payoff externalities are better suited for modeling settings such as resource competition (e.g., \[3\]). Thus, our work is complementary to the work in MARL but more closely aligns with the work in social learning and information design. We more specifically discuss the works mentioned by the reviewer (\[2\]-\[6\]) below.
> > > >
> > > > 1. Missing Planner: \[2\]-\[5\] study the emergence of system dynamics (e.g., stability) from decentralized interactions. These models lack a planner (1) attempting to steer the equilibrium. Consequently, they describe how herding emerges or the impact of herding on system dynamics (e.g., stability as in \[4\]) but do not provide the framework required to design information to manage these phenomena. That is, they do not address the *planner’s* problem.
> > > > 2. Lack of inter-agent learning: \[6\] considers a setting with a single receiver. Thus, their model cannot capture the social dynamics which are a central concern of our work. We hence maintain that \[6\] does not provide a model to study the effect of social learning in information design problems. Nonetheless, this is a related work of interest and we have now included it in our related work.
> > > > 3. Payoff vs. Information Externalities: Another key distinction is the mechanism of social learning considered. Works such as \[3\]-\[5\] model social learning via payoff externalities, but studying the influence of information design requires models that capture information externalities (as is done in our work).
> > > >
> > > > ## LLMs as Human Simulators
> > > >
> > > > While we maintain that the value of our findings does not rely upon the accuracy of LLM-human simulation, we agree that the limitations of LLMs as human simulators are important and worthy of discussion. As per the reviewer’s suggestion, we have mentioned this explicitly in our conclusion.

---

> > > > > ### Author Response · Authors · 2025-12-03
> > > > >
> > > > > ## References
> > > > >
> > > > > \[1\]	S. Bikhchandani, D. Hirshleifer, O. Tamuz, and I. Welch, “Information Cascades and Social Learning,” Journal of Economic Literature, vol. 62, no. 3, pp. 1040–1093, Sept. 2024, doi: 10.1257/jel.20241472.
> > > > > \[2\]	Z. Yang *et al.*, “OASIS: Open Agents Social Interaction Simulations on a Large Scale,” Oct. 2024, Accessed: Nov. 27, 2025\. \[Online\]. Available: https://openreview.net/forum?id=JBzTculaVV
> > > > > \[3\]	S.-P. Zhang, J.-Q. Dong, L. Liu, Z.-G. Huang, L. Huang, and Y.-C. Lai, “Artificial intelligence meets minority game: toward optimal resource allocation,” *Phys. Rev. E*, vol. 99, no. 3, p. 032302, Mar. 2019, doi: 10.1103/PhysRevE.99.032302.
> > > > > \[4\]	J. Lussange, S. Vrizzi, S. Palminteri, and B. Gutkin, “Mesoscale effects of trader learning behaviors in financial markets: A multi-agent reinforcement learning study,” *PLOS ONE*, vol. 19, no. 4, p. e0301141, Apr. 2024, doi: 10.1371/journal.pone.0301141.
> > > > > \[5\]	E. Sangati, A. Chang, and K. Doya, “Social Amplification Can Help Solve the Credit Assignment Problem in Collective Learning,” presented at the ALIFE 2025: Ciphers of Life: Proceedings of the Artificial Life Conference 2025, MIT Press, Oct. 2025\. doi: 10.1162/ISAL.a.862.
> > > > > \[6\]	Y. Lin, W. Li, H. Zha, and B. Wang, “Information Design in Multi-Agent Reinforcement Learning,” *Adv. Neural Inf. Process. Syst.*, vol. 36, pp. 25584–25597, Dec. 2023\.

---

### Official Review · Reviewer_n8ZY · 2025-11-04

**Soundness:** 4
**Presentation:** 4
**Contribution:** 4
**Rating:** 8
**Confidence:** 2

**Summary:**

This paper studies how a social planner can intervene on signal strength in the information cascade model to induce both altruistic and biased outcomes. The authors derive optimal policies for the planner in both cases, in both the setting where the social learning aspect is ignored and when the social learning aspect is taken into account. They then study these dynamics with LLMs posing as the agents, the social planner, and a belief oracle.

**Strengths:**

The main contributions of this work are theoretical results for intervening on signal strength and an interesting experimental setup to evaluate the theoretical results. I am not aware of this particular style of intervention, and I am excited to see clear and insightful theoretical work in this vein. The paper is also clearly written and laid-out.

**Weaknesses:**

Some small recommendations:
1. include more of the experimental setup in the main body. It would just help to clarify how exactly you set this up with prompting etc.
2. the theorems may benefit from diagrams to show what the policies are in the different regimes.
3. worth having one extra line about computation about $\tilde{b}_i$ in the main body as otherwise it is quite mysterious to the reader since it is a crucial point.

**Questions:**

-

---

> ### Author Response · Authors · 2025-11-22
> **Author Response**
>
> We thank the reviewer for their positive rating and remarks on the novelty and insight of our paper. We are also grateful for their recommendations which helped us revise our submission for greater clarity. We note specific edits corresponding to their comments below:
>
> ## Experimental setup
>
> Following the reviewer's suggestion, we have now augmented the start of Section 6 with greater detail from Appendix E on the experimental setup. We believe this makes the main body of the paper more self-contained.
>
> ## Theorem clarity
>
> We show examples of the optimal altruistic and biased policies in figure 2a. We have added references after the statements of theorems 3 and 5 to direct readers toward these examples.
>
> ## Private posterior belief update
>
> The reviewer is correct in pointing out the importance of $\\tilde b\_i$ to the social learning process. We have added its explicit update equation as Equation 1 of our paper.

---

### Official Review · Reviewer_LWVF · 2025-11-04

**Soundness:** 3
**Presentation:** 2
**Contribution:** 4
**Rating:** 8
**Confidence:** 3

**Summary:**

This paper revisits sequential social learning (Bikhchandani et al.) and adds a centralized planner who, at each step, chooses the precision of the next agent’s private signal. Agents best respond given the public belief and the stated precision. The planner’s problem (altruistic vs. biased objective) is cast as an infinite‑horizon discounted MDP in the public belief state. A key theoretical contribution is proving that the altruistic optimal value function $V^*_A$ is convex in belief, which underpins a threshold‑type policy structure; the paper also reports LLM‑based simulations.

**Strengths:**

The convexity proof of $V^*_A$ is neat.  It gives a clear policy structure and may be useful for other related problems, e.g., dynamic information acquisition, and sequential contract design.

**Weaknesses:**

- The model is a little too restrictive.  It is not clear if their result can go beyond symmetric binary signal setting.
- The LLM experiment require more cares.  Figure 1 is misleading, as the value of prior and posterior are self-reported by LLMs.  Similarly, they should validate the strength of the signal from the oracle agent, e.g., asking another LLM to guess q_i.  I do not feel comfortable comparing those values with Bayesian models.


### Miner issue
- The theorem statement can be clearer.  For instance, though Theorem 1 only works for $\delta = 1$, Theorem 2 should hold for all discount factors $\delta<1$.
- Equation (28) should depend on prefix of the trajectory but $P^*_\lambda$ seems to the probability of whole trajectory.
- You may try to simplify the proof by using dynamic programming (Bellman operator) or coupling.

**Questions:**

Instead of a symmetric binary signal and 01 loss, how is the result generalized to a general signal and a loss function?

---

> ### Author Response · Authors · 2025-11-22
> **Author Response Part I**
>
> We thank the reviewer for the positive rating and appreciation of our convexity proof of the value function. We agree that there may be applications for this result in other areas and are excited to explore this possibility in future work. We also greatly appreciate the reviewer’s time and thoughtful feedback. Below we've included revisions, clarifications, and additional experiments which we hope address the reviewer's concerns:
>
> ## Model restrictions
>
> The reviewer makes a great point \-- it is worthwhile to discuss the ways in which we might strip away the assumptions of our model. Below we discuss some generalizations of our problem and their feasibility. We have also incorporated these generalizations in our conclusion.
>
> ### Beyond the Binary Case
>
> We consider a binary setting as in classical works in sequential social learning (e.g., [1]). This is primarily for ease of exposition. As in [2], it is possible to relax this to a finite setting, but there is "significant algebraic cost". For $k$ states each with their own respective optimal action, the belief would be a point on the $k-1$-dimensional simplex and each action would be optimal within a convex region of the simplex.
>
> ### More General Signal Structures
>
> The most significant challenge in considering more general signal structures is ensuring that the convexity result of Theorem 2 still holds. We currently prove the result for the binary and symmetric case where both the state (public belief) and action (chosen precision) are scalar. It is not clear whether the neat threshold structure of our results (e.g., the simple form of the optimal altruistic policy in Theorem 3\) would still hold in higher dimensional settings. Finding the most general class of signal structures for which such a result holds is an interesting problem for future study.
>
> ### More General Loss Functions
>
> With discrete actions, any expected utility maximizing agent's decision reduces to choosing convex decision regions over the belief simplex. A change in loss functions then amounts to shifting the corresponding decision boundaries. In the binary case, the 0-1 loss function we assume has a decision threshold at 0.5, i.e., an agent's optimal choice is $G$ if their posterior belief is above 0.5. For a different loss function this specific threshold may change; however, so long as agent preferences are known to all, future agents and the planner will still be able to properly infer information from actions and maintain the public belief. Thus, the thresholds and boundaries of our analysis may shift, but we do not expect a change in the qualitative results. In a related generalization, we have extended our results for the altruistic planner to the case of two agent types with opposing preferences (i.e., loss functions) in Appendix D.
>
> ## References
> [1]	S. Bikhchandani, D. Hirshleifer, and I. Welch, “A Theory of Fads, Fashion, Custom, and Cultural Change as Informational Cascades,” https://doi.org/10.1086/261849, vol. 100, no. 5, pp. 992–1026, 1992, doi: 10.1086/261849.
>
> [2]	L. Smith and P. Sørensen, “Pathological outcomes of observational learning,” Econometrica, vol. 68, no. 2, pp. 371–398, 2000, doi: 10.1111/1468-0262.00113.

---

> ### Author Response · Authors · 2025-11-22
> **Author Response Part II**
>
> ## LLM Belief and Oracle Validation
>
> This is a great point. To clarify, in the experiments used for Figure 1b of our original submission, only the posterior was self-reported by the LLM while the prior was provided so that we could study belief updating for all possible prior beliefs. We now clarify this in the caption of the figure. Following the reviewer's question, we now provide further evidence to substantiate that the posterior reported by the LLM is reliable. In particular, we have conducted further experiments summarized below with results added to Appendix E.3. In these experiments, we utilize the LLMs as rational and utility maximizing, but not necessarily Bayesian, agents. The goal of these experiments is to demonstrate that the model’s self-reported posterior genuinely corresponds to the belief that governs its choices in state-contingent decision problems.
>
> First, we provide the LLM with a prior and a new signal, obtain its self-reported posterior belief, and then ask it to make a binary “buy/not-buy’’ decision that yields unit reward if and only if the true state is good. A rational agent should choose “buy’’ exactly when its subjective belief exceeds 0.5. As shown in Figure 4a of Appendix E.3, the LLM’s decisions align sharply with this threshold: it overwhelmingly chooses “buy’’ when its reported posterior exceeds 0.5 and “not buy’’ otherwise.
>
> Next, we elicit the model’s willingness to pay for a payoff that is contingent on the true state. For a range of prices $c$, we ask whether the agent would pay $c$ in exchange for receiving unit utility if the state is good. A rational agent with belief $p$ should accept exactly when $c\<p$. Figure 4b plots the fraction of "buy" decisions across posterior–price pairs, showing a clear separation: acceptance rates are high when $c\<p$ and low when $c\>p$, precisely as expected from a utility-maximizing decision maker whose belief is $p$. Both experiments demonstrate that the self reported posterior is close to the inherent belief of the model as used in decision making.
>
> Similarly, we validate the performance of the oracle as suggested by the reviewer. After the oracle generates a message of specified precision $q$, we ask another LLM to form an estimate $\\hat q$ of the message's precision. Figure 2 below shows a histogram of $q-\\hat q$. We do this for $q\\in\\{0.5, 0.6, 0.7, 0.8, 0.9, 1.0\\}$ on 100 programmatically generated agent profiles with 10 trials for each combination for a total of 6,000 trials. The mean absolute error $|q-\\hat q|$ was just 0.025. This indicates that the realized precision of the oracle’s message does closely follow the requested precision.
>
> ## Theorem Statement Clarity
>
> We define the optimal myopic policy $\\pi^0\_A$ (characterized in Theorem 1\) as the $\\arg\\sup$ of the instantaneous reward $r\_A(b,q)$ (Eqn. 6). Thus, $\\pi^0\_A$, and subsequently Theorem 1, are independent of the discount factor $\\delta$ although they can be interpreted as applying discount factor $\\delta=0$ from the value function defined in Eqn. 5\.
>
> The reviewer is correct in asserting that Theorem 2 holds for $\\delta\\in\[0,1)$. This assumption was stated immediately before we define the value function in Eqn. 5 and holds throughout the paper. We have now clarified this immediately after the definition of the value function.
>
> ## Theorem 2
>
> On Eqn. 28, yes, the reviewer is correct. Equation 28 contained an off-by-one error in the time index which has been corrected. We thank the reviewer for their attention to detail\!
>
> On ways to simplify the proof, it is not immediately clear if such classical arguments can be used to cut down the proof given the challenges we discuss in the paper (in particular the nature of the dependence of agents’ actions on the public belief). We aim to investigate this possibility further as we look at generalizations beyond the special binary symmetric signaling that is assumed in this work.

---

### Author Response · Authors · 2025-12-03
**Summary of Discussion Period and Thanks to the Reviewers**

We thank the reviewers for their time and thoughtful feedback. We are encouraged by the positive remarks highlighting the novelty of our framework, the rigor of our theoretical results, and the timely integration of theoretical analysis with LLM-based simulations.

We have carefully addressed the reviewers' comments, leading to substantial improvements in the manuscript. Below is a summary of the key developments and revisions:

1. **Clarification of Positioning and Novelty (Reviewer gvhy):**

We have significantly **augmented the literature review** (Section 2\) to explicitly distinguish our work from those in online persuasion and RL. We clarified that while MARL often studies the *emergence* of herding via payoff externalities, our work focuses on the *control* of social learning via information externalities—a distinction critical for studying our motivating applications. Following these clarifications and the inclusion of suggested references, **Reviewer gvhy acknowledged our initial rebuttal and raised their score from 2 to 4**. We have since further clarified the distinction between our work and specific papers mentioned by the reviewer.

2. **Validation of Experimental Methodology (Reviewer LWVF):**

To address concerns regarding the reliability of LLM self-reported beliefs, we **conducted new experiments** (added to Appendix E.3). These results demonstrate that the LLM's reported posterior aligns sharply with its decision-making in state-contingent tasks (willingness-to-pay and binary choice), validating our experimental proxy. We also validated the precision of the LLM Oracle.

3. **Expanded Discussion on Generalizations (Reviewers LWVF, v6hQ):**

We have **expanded the conclusion and Appendix** to discuss the feasibility of relaxing model assumptions, including non-binary states, general signal structures, and heterogeneous agent preferences.

4. **Presentation and Clarity (Reviewer n8ZY):**

We have **improved the exposition** of the experimental setup in Section 6 and added explicit update equations for private beliefs (Equation 1\) to make the main body more self-contained.

We believe these revisions have strengthened our paper considerably and are grateful to the reviewers for their valuable feedback.

---

### Meta-Review · Area_Chair_fcku · 2026-01-05

**Summary:**

Reviewers found the (theoretical) results interesting and potentially can be used for other studies. Suggestions/concerns were raised about restrictiveness of the model, and some other minor comments on presentation and clarity. Overall, this looks like an exciting work to be included in the program.

**Reviewer Concerns:**

see above

**Reviewer Scores:**

NA

---

### Decision · Program_Chairs · 2026-01-26

Accept (Oral)